# The space of integrable systems from generalised $T\bar{T}$-deformations

**Benjamin Doyon[1,2], Joseph Durnin[1] and Takato Yoshimura[3⋆]**

**1** Department of Mathematics, King's College London,
Strand, London WC2R 2LS, United Kingdom
**2** Department of Physics, Tokyo Institute of Technology,
Ookayama 2-12-1, Tokyo 152-8551, Japan

## Abstract

We introduce an extension of the generalised $T\bar{T}$-deformation described by Smirnov-Zamolodchikov, to include the complete set of extensive charges. We show that this gives deformations of S-matrices beyond CDD factors, generating arbitrary functional dependence on momenta. We further derive from basic principles of statistical mechanics the flow equations for the free energy and all free energy fluxes. From this follows, without invoking the microscopic Bethe ansatz or other methods from integrability, that the thermodynamics of the deformed models are described by the integral equations of the thermodynamic Bethe-Ansatz, and that the exact average currents take the form expected from generalised hydrodynamics, both in the classical and quantum realms.

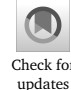 Check for updates

doi:10.21468/SciPostPhys.13.3.072

# 1 Introduction

Much progress is made in the study of many-body systems by the analysis of special classes of theories where exact properties can be extracted. The main historical example is that of integrable systems, which occur in a variety of physical setups, from classical gases to quantum field theories (QFT). Quite surprisingly, regardless of the specific setup, these systems share many properties and common descriptions, such as the factorisation of scattering amplitudes into two-body scattering phases or matrices [1,2], the thermodynamic Bethe ansatz (TBA) [3, 4] and generalised hydrodynamics (GHD) [5, 6]. However, progressing beyond special examples to an understanding of the full space of integrable many-body systems, and uncovering the origin of descriptions common to all such systems, remain challenging problems.

Recently a new, and rather broad class of deformations of integrable systems has come to the fore, enlarging the space of known models. These are the so-called $T\bar{T}$-deformations, where relativistic QFTs are deformed by the determinant of the stress-energy tensor $T_{\mu\nu}$ [7–9]. The principles underlying $T\bar{T}$-deformations have been extended to non-relativistic systems [10–13], where they are best expressed as charge-current deformations, as well as to relativistic systems possessing higher-spin charges [9, 14–17]; the nomenclature $T\bar{T}$-deformation refers to all of these. It is found in particular that higher-spin $T\bar{T}$-deformations lead to a large class of two-body scattering phases, differing in their CDD factors [8, 9, 17].

A considerable amount of physical insights into the (usual) $T\bar{T}$-deformation has been gained so far. For instance, it has been shown that the $T\bar{T}$-deformation can be thought of as a coupling of the original theory to random geometry [18]. In a somewhat similar vein, one can also argue that $T\bar{T}$-deforming a two-dimensional QFT is the same as coupling the theory to a two-dimensional topological gravity [19]. Interestingly, there is a holographic interpretation of the $T\bar{T}$-deformation provided that the original theory is a conformal field theory with the holographic dual [20]. Another important interpretation is that the usual $T\bar{T}$-deformation is equivalent to a state-dependent coordinate change, which is controlled by the local stress-energy tensor [11, 21].

The notion of state-dependent change of coordinates was in fact introduced earlier, in the context of generalised hydrodynamics, where it was found to map the hydrodynamics of free particles to that of interacting integrable models [22]. It generalises the change-of-width map

that has long been used for hard rods, and has an algebraic analogue in the $q$-boson chain [23]. It is this viewpoint that led to yet another way of looking at the $T\bar{T}$-deformation, at the basis for the present paper: that the non-relativistic version of the original $T\bar{T}$-deformation corresponds to a change in particle widths [12, 13, 24].

In the context of GHD, it was in fact understood that assigning particle widths that *depend on the scattering momenta* [25] reproduces the thermodynamics and Euler hydrodynamics of integrable systems with arbitrary two-body scattering displacements. The resulting classical "flea gas" [25], however, does not have a well-behaved Hamiltonian microscopic dynamics. But the insight that changes of particle widths are connected to $T\bar{T}$-deformations is very suggestive: it is natural to think that $T\bar{T}$-deformations offer one possible route to constructing a universal, Hamiltonian formulation for a large space of factorised-scattering integrable systems, with arbitrary two-body scattering amplitudes.

In this paper, motivated by this insight and the work by Smirnov and Zamolodchikov [9], we present a new family of deformations which produce integrable systems, whose S-matrix has arbitrary momentum dependence. This is achieved by considering not only the local conserved charges of the initial theory, but the complete space of extensive conserved charges. This forms a Hilbert space which also contains quasi-local charges [26–28], and a complete basis can be labelled by the momenta of asymptotic particles [29–31]. The consideration of completeness and of an appropriate basis has played an important role in the understanding of generalised thermalisation [26, 32] and of the hydrodynamic projection principle [28]. Here, we show that completeness also plays a crucial role in $T\bar{T}$-deformations.

We use a general formalism for statistical mechanics that accounts for an arbitrary number of extensive conserved quantities, as is used in the context of generalised hydrodynamics (see e.g. [33]). The formalism, derivation and results are valid for arbitrary many-body systems, be they quantum or classical, integrable or not, relativistic or not.

Within this formalism, we also obtain a new, model-independent derivation of the flow equations that describe how thermodynamic quantities change under $T\bar{T}$-deformations. We obtain the flow equations not only for the free energy, as is traditionally done, but also for the free energy fluxes [5, 34], which generate average currents and are related to entropy fluxes. This extends previous results, and in particular the result [16] concerning flows under Smirnov-Zamolochikov deformations, as it gives the flow equations under arbitrary $T\bar{T}$ deformations, including the new deformations discussed here associated to quasi-local charges, and it provides the flow for the free energy flux as well. It also shows, by contrast to previous derivations, how the flow equations are based solely on *general properties of many-body systems*, and not on specific structures of integrability or quantum field theory. Applying this to integrable models, we show that the general flow equations are solved by the TBA formulation for the free energy and free energy fluxes. This provides a new, universal derivation of the TBA and its extension to the exact average of currents, both for quantum and classical models, without the use of the Bethe ansatz or other methods from integrability. We believe this sheds light on why the TBA and GHD are such widely applicable frameworks.

In this paper, we concentrate on the general and universal aspects of the construction. Explicit examples of deformations that go beyond what has been done in the pre-existing literature would require both explicit forms of quasi-local charges, which are known only in few models, and an independent analysis of the resulting theory with this deformation, which, because of the quasi-local nature of the deformation, may require going beyond standard integrability methods.

We believe the importance of the complete space of extensive conserved charges, and the strength of the general formalism of statistical mechanics that accounts for these, has not been appreciated enough beyond the area of non-equilibrium systems, where these concepts originated. In particular, these concepts have been overlooked in studies of $T\bar{T}$-deformations

until now. This paper provides one of the first applications to the study of general structures of many-body systems.

The paper is organised as follows: we first recall how the "conventioinal" $T\bar{T}$- deformations as well as other integrable deformations are defined in terms of local charges in Sec. 2. This will also motivate us to introduce the notion of quasi-local charges in order to expand the realm of deformations, which is achived in Sec. 3. We then explain how the thermodynamics of generalised $T\bar{T}$-deformed theories can be accessed by flow equations for free energy and free energy fluxes in Sec. 4, where the derivation of the equation for the free energy fluxes is consigned to Appendix C and D. In the case of integrable systems, it turns out the solution of the equations correctly reproduces the results of TBA, which serves as a novel derivation of TBA without resorting to Bethe ansatz machienery and reported in Sec. 5. We also briefly discuss the structure of the $S$-matrix obtained from the deformations, and compare with the properties that are expected to satisfy in integrable QFTs in Sec. 6. We then finally conclude in Conclusion 7.

## 2 Integrable deformations by (quasi-)local charges

Let us take integrable QFTs (IQFTs) as an example, though the concepts apply equally well to any many-body integrable system with sufficiently local interactions. There are infinitely-many conservation laws, corresponding to local charges $Q_i = \int_{\mathbb{R}} dx\, q_i(x)$ in involution $[Q_i, Q_j] = 0$, where $q_i(x)$ is a local density. In IQFT, imposing strict locality of the conserved charges restricts its spin, see e.g. [2]; for instance, in the sinh-Gordon model, the spins of the local charges must be odd. In this paper, we focus on systems on infinite volume, although finite volumes are considered in intermediate steps.

We are enquiring about deformation of the Hamiltonian $\delta H$ which preserve integrability. We denote this as $\delta H \in T\Sigma^{\mathrm{Int}}$. The notation suggests the existence of a manifold $\Sigma^{\mathrm{Int}}$ of integrable models, whose tangent space $T\Sigma^{\mathrm{Int}}$, at a given model, is the space of its integrability-preserving deformations.

One simple family of deformations is $\delta H = \delta\eta\, Q_i$ for some $i$; this is a redefinition of time, thus of the dispersion relation. It trivially preserves the full integrable hierarchy and hence the two-body scattering, and so, certainly, $\delta H \in T\Sigma^{\mathrm{Int}}$. Another family of deformations is $\delta H = \delta\kappa\, J_i = \delta\kappa \int_{\mathbb{R}} dx\, j_i(x)$ in terms of the currents, $\partial_t q_i + \partial_x j_i = 0$. These are generated by the boost $B_i = \int_{\mathbb{R}} dx\, x q_i(x)$, as $J_i = \mathrm{i}[H, B_i]$. Because these deformations form a Hamiltonian flow, they preserve integrability [35–38], and thus again $\delta H \in T\Sigma^{\mathrm{Int}}$. The momentum flows as $\delta P = \delta\kappa\, \mathrm{i}[P, B_i] = \delta\kappa\, Q_i$, thus this is a redefinition of the spatial direction by flows in the integrable hierarchy. With respect to the new spatial direction, the two-body scattering is unchanged, as noted in [35–37].

The integrable deformations introduced by Smirnov and Zamolodchikov [9] are a "bilinear mix" of these two types of deformations. Namely, the deformation

$$\delta H = \delta\lambda \int_0^R dx\, (q_i(x-\epsilon)j_j(x) - j_i(x-\epsilon)q_j(x)), \tag{1}$$

generates a flow of new IQFTs, $\delta H \in T\Sigma^{\mathrm{Int}}$. The infinitesimal parameter $\epsilon$ ensures point-splitting and $R$ is the system size, where a periodic boundary condition is imposed; as mentioned, we will be interested in the limit $R \to \infty$. Deformations (1) generate modifications of the two-body scattering matrix by so-called CDD factors. The form of CDD factor is determined by the spins of the charges $Q_i, Q_j$, and the restriction imposed by locality on the spins of the charge guarantee that the factor produced is indeed of CDD type for the model

in question. The flow equations, describing how the free energy and other physical quantities are deformed, were obtained in [16] using QFT techniques.

We make the crucial remark that the charges appearing in (1) do not have to be local in the usual sense, but can also be quasi-local. Quasi-local charges have densities $q(x)$ not strictly supported on the point $x$, but on extended regions around $x$, with sufficiently decaying operator norm. These charges are easy to construct in free field theories using bilinear forms, and appear in the description of non-equilibrium steady states [39] (see also Appendix A for the example of Majorana fermions). They naturally arise from the transfer matrix formalism in models with non-diagonal scattering [26,40], such as the XXZ spin-1/2 chain [26], and in a scaling limit of lattice theories [41]. Local and quasi-local charges are part of the Hilbert space of extensive conserved charges [27], where the inner product is $\lim_{R\to\infty} R^{-1}\big[\langle Q_i Q_j\rangle - \langle Q_i\rangle\langle Q_j\rangle\big]$, and is finite. Extensive charges are relevant; for instance, studies in many-body systems [26, 33] indicate that the only required property for (Hermitian) conserved charges to play a role in the non-equilibrium physics of relaxation and hydrodynamics is extensivity.

Thus, in general, it is natural to ask about deformations of the type (1), but which include a sum over a *complete basis* in the space of extensive charges,

$$\delta H = \sum_{ij} \delta\lambda^{ij} \int_0^R \mathrm{d}x\, (q_i(x-\epsilon)j_j(x) - j_i(x-\epsilon)q_j(x)). \tag{2}$$

In integrable systems in particular, this concept points to a more judiciously chosen set of charges: the $Q_\theta$, $\theta \in \mathbb{R}$, measuring the density of asymptotic particles of rapidity $\theta$. The rapidities, labelling the asymptotic particles' momenta, are preserved by the scattering map, and are therefore a particular choice of action variables. (We assume the presence of a single particle species for simplicity.) These charges exist only in the limit $R \to \infty$, but in integrable systems, they have smoothly connected analogues at finite $R$, with $\theta$ taking discrete values, by the Bethe ansatz or the inverse scattering methods. They constitute a complete "scattering" basis for the extensive conserved charges: much like scattering states in quantum mechanics, each $Q_\theta$ is not strictly extensive (normalisable), but is a limit of an extensive conserved charge obtained by a wave packet construction (see e.g. [29–31]). They also constitute a basis for the generalised Gibbs ensemble (GGE), which we shall employ in this paper: $\varrho = Z^{-1} \exp\big(-\int_{\mathbb{R}} \mathrm{d}\theta\, \beta^\theta Q_\theta\big)$, where $\mathrm{Tr}\,\varrho = 1$ and $\beta^\theta$ are the rapidity-dependent Lagrange multipliers [29,42]. The usual local charges take the form $Q_i = \int_{\mathbb{R}} \mathrm{d}\theta\, h_i(\theta)Q_\theta$ where $i$ labels, for example, the allowed spins. One may choose $h_i(\theta)$ so that $Q_i$ form a complete, discrete basis; but it is important to emphasise that in general, the local charges do not form such a basis (that is, quasi-local charges are required within the basis set).

As the $Q_\theta$ form a basis for the space of extensive charges and for expressing GGEs, it is natural to suggest that they can be used to obtain deformations that span, along with the simpler $\eta$- and $\kappa$-deformations above, the tangent space $T\Sigma^{\mathrm{Int}}$. We now argue that this is indeed so.

## 3 Generalised Boost and $T\bar{T}$-deformations

For free-particle models, where the scattering shifts vanish modulo particle statistics, it is trivial to construct $Q_\theta$. We now show that such models can be continuously deformed into interacting integrable models, parametrised by a deformation parameter corresponding to the scattering phase shift.

As they are extensive (in the sense discussed above), the charges $Q_\theta$ for each rapidity $\theta \in \mathbb{R}$ have associated charge densities $q_\theta(x)$ and generalised currents [5, 34, 43–47] $j_{\theta\phi}(x)$, where

$Q_\theta = \int dx\, q_\theta(x)$ and the continuity equation reads

$$i[Q_\theta, q_\phi(x)] + \partial_x j_{\theta\phi}(x) = 0. \tag{3}$$

Suppose the system contains a single species of particle, which may be quantum or classical, and has Hamiltonian and momentum

$$H = \int_{\mathbb{R}} d\theta\, E(\theta) Q_\theta, \quad P = \int_{\mathbb{R}} d\theta\, p(\theta) Q_\theta. \tag{4}$$

Here the energy and momentum $E(\theta), p(\theta)$ characterise the dispersion relation, e.g. $E(\theta) = \theta^2/2$, $p(\theta) = \theta$ and $\cosh(\theta)$, $\sinh(\theta)$ for Galilean and relativistic systems, respectively.

The $\eta$-deformation can be expressed as $\delta H = \int_{\mathbb{R}} d\phi\, \delta\eta_\phi Q_\phi$, and corresponds to a different choice of energy $E_\eta(\theta) = E(\theta) + \eta_\theta$, thus changing the dispersion relation. Here we make an important observation: although the $\eta$-deformation is rather simple and well-known, restricting to local charges as is conventionally done, $\eta_\theta = \sum_{i:\text{local}} c_i h_i(\theta)$, would restrict the possible deformations of the dispersion relation. For instance, in the sinh-Gordon model, the deformation with local charges is of the form $E_\eta(\theta) = E(\theta) + \sum_{i\in\mathbb{Z}} c_i e^{(2i+1)\theta}$, and, although the parameters $c_i$ are arbitrary, the sum cannot generate an arbitrary function. By contrast, we see that with the complete basis, $\eta_\theta$ is an arbitrary function of $\theta$ – up to weak restrictions such as its growth at large $\theta$, which guarantee that $\int_{\mathbb{R}} d\phi\, \delta\eta_\phi Q_\phi$ is quasi-local – and we may change the dispersion relation arbitrarily.

Under the $\kappa$-deformation, the charges are deformed as $\delta Q_\theta = \int_{\mathbb{R}} d\phi\, \delta\kappa_\phi J_{\theta\phi}$, as generated by the boost operator, $\frac{\delta Q_\theta}{\delta\kappa_\phi} = i[Q_\theta, B_\phi]$. Using $\int_{\mathbb{R}} d\theta\, p(\theta) J_{\theta\phi} = Q_\phi$, which follows from (3) and (4), the momentum becomes $p_\kappa(\theta) = p(\theta) + \kappa_\theta$. Again, we obtain in this way an arbitrary momentum re-parametrisation; other eigenvalues also transform.

More far-reaching is the use of a complete set of charges in the $\lambda$-deformation. In order to fully understand it, we consider the analogue of (1), or rather the more general (2), not only for the Hamiltonian, but for all conserved quantities. Further, as was done before in studies of $T\bar{T}$-deformations, a careful consideration of the generalised form of the deformation requires us to consider finite volumes, where we take periodic conditions. Thus we write

$$\delta Q_\gamma = \int_{\mathbb{R}} d\theta \int_{-\infty}^{\theta} d\phi\, \delta\lambda_{\theta\phi} \mathcal{O}_{\gamma\theta\phi}, \tag{5}$$

where

$$\mathcal{O}_{\gamma\theta\phi} = \int_0^R dx\, \left( q_\theta(x-\epsilon) j_{\gamma\phi}(x) - j_{\gamma\theta}(x-\epsilon) q_\phi(x) \right). \tag{6}$$

We shall call this class of deformations "generalised $T\bar{T}$-deformations" in this paper.

The deformation (5) with (6) must be understood appropriately. As in all studies of $T\bar{T}$-deformations, one considers a flow generated by these deformation equation. That is, for any given $\lambda_{\theta\phi}$, there is a $\lambda$-dependent set of conserved charges $Q_\gamma$, and their infinitesimal deformation under infinitesimal changes $\lambda_{\theta\phi} \to \lambda_{\theta\phi} + \delta\lambda_{\theta\phi}$ is given by (5) where the operator $\mathcal{O}_{\gamma\theta\phi}$ involves the densities and currents *of the $\lambda$-dependent set of conserved charges $Q_\gamma$*. The most powerful way of dealing with the flow is that pointed out in [37, 48], where the standard $T\bar{T}$-deformations are generated by appropriate bilinear operators. It is a simple matter to observe that the deformation (5) with (6) is likewise generated by the following bilinear operator:

$$X_{\theta\phi} = -\int_0^R dy \int_0^{y-\epsilon} dx\, q_\theta(x) q_\phi(y), \; \theta \geq \phi, \tag{7}$$

in the sense that

$$\frac{\delta Q_\gamma}{\delta \lambda_{\theta\phi}} = i[X_{\theta\phi}, Q_\gamma] + Q_\theta j_{\gamma\phi}(0) - j_{\gamma\theta}(0)Q_\phi. \tag{8}$$

For convenience we take $\lambda_{\theta\phi}$ to be 0 when $\theta < \phi$, as it will be seen that only the antisymmetric part contributes.

Notice that while the first term on the RHS of (8) does not change the spectrum of the system, the second and the third terms do. This transformation preserves the integrability by a simple extension of the results of [9, 12], and also by considering the standard arguments for the properties of integrability [49]. Thus the S-matrix of the deformed model preserves its factorised structure. We now evaluate the deformation of the S-matrix, for both quantum and classical systems, by considering the 2-particle scattering problem.

For a classical system of $N$ free particles, we may use the methods of [13]. We observe that $q_\theta(x) = \sum_{i=1}^{N} \delta(x - x_i)\delta(\theta - \theta(p_i))$, where $x_i$, $p_i$ are the canonical coordinates for $i = 1, \cdots, N$. The deformed charge density is simply given by the same expression with deformed coordinates, which stay canonical. Thus it is sufficient to calculate the action of $X_{\theta\phi}$ in the free model, and one obtains the solution for all $\lambda_{\theta\phi}$, see Appendix B. This yields the following deformed S-matrix:

$$S_\lambda(\theta, \phi) = e^{-i(\lambda_{\theta\phi} - \lambda_{\phi\theta})} S(\theta, \phi), \tag{9}$$

where $S(\theta, \phi)$ is the S-matrix of the undeformed model. The same expression is obtained in quantum systems by consideration of the spatially ordered 2-particle wavefunctions. Thus we see that, as claimed, by a suitable choice of $\lambda_{\theta\phi}$, an arbitrary 2-particle S-matrix can be obtained. The only condition is the anti-symmetry of the scattering phase, but this simply means that unitarity of the scattering matrix is preserved.

The result above is expected to be general for many-body systems; it is expressed in terms of a quantum scattering matrix, but (as is clear from Appendix B) the result applies as well to classical systems. In the many-body theory of scattering, a number of properties of the scattering matrix are expected to be satisfied; in QFT this is made quite precise, as encoded in analyticity properties and their relations to the emergence of bound states. We discuss these aspects below. First, however, let us discuss the thermodynamics and the associated flow equations.

## 4 Flow Equations

We now describe the thermodynamics and the equations of state (the average currents) by writing down the explicit flow equations for both the *free energy density f* as well as for the *free energy fluxes $g_\theta$* (see below). Although we discuss integrable models, the flow equations can be written in the complete generality of many-body systems with an arbitrary number of conserved charges. We provide the general form of the flow equations, and a general derivation purely based on general principles of many-body physics, in Appendices C and D.

The thermodynamic state is a GGE, specified by an infinite set of parameters $\beta^\theta$, which are conjugate to the conserved quantities $Q_\theta$. Below we reproduce the proof, in our setting, that the fact that the charges are conserved and in involution in the free system, implies that their deformations also are conserved and in involution for the deformed system; thus at all values of the deformation, we have a GGE.

### 4.1 Generalised conservation laws

We may associate a generalised time $t_j$ to each conserved quantity $Q_j$ by a generalised Heisenberg equation, $\partial_{t_j} \mathcal{O} = i[Q_j, \mathcal{O}]$, where $\mathcal{O}$ is any local observable. This construction

is valid if the charges are in involution. Recall the generalised currents defined in (3); here we consider generalised currents in the generic basis $j_{ik}$. Then, a generalised conservation law holds,

$$\partial_{t_i} j_{jk} = \partial_{t_j} j_{ik} \, . \tag{10}$$

Indeed,

$$\partial_{t_i} \partial_x j_{jk} = -\partial_{t_i} \partial_{t_j} q_k = -\partial_{t_j} \partial_{t_i} q_k = \partial_{t_j} \partial_x j_{ik} \, , \tag{11}$$

and this implies that $\partial_{t_i} j_{jk} - \partial_{t_j} j_{ik}$ is independent of $x$, and hence proportional to the identity operator. As this expression vanishes in any equilibrium state by virtue of the Gibbs form $\rho \propto e^{-\beta^i Q_i}$, the coefficient of proportionality is 0 and Eq. (10) holds. This expression contains the usual charge conservation equation, as generally $H = h^i Q_i$ for some coefficients $h^i$, and thus $\mathrm{i}[H, q_k] = h^i \partial_{t_i} q_k = -h^i \partial_x j_{ik}$. This allows us to identify the physical currents $j_k = h^i j_{ik}$.

## 4.2 Commutativity under the flow

Now, recall that charges $Q_i$ are deformed as (5) with (6), in the generic basis:

$$\delta Q_i = \int_{\mathbb{R}} \mathrm{d}\theta \int_{-\infty}^{\theta} \mathrm{d}\eta \, \delta\lambda_{\theta\eta} \mathcal{O}_{i\theta\eta} \, , \tag{12}$$

with

$$\mathcal{O}_{i\theta\eta} = \int_0^R \mathrm{d}x (q_\theta(x-\epsilon) j_{i\eta}(x) - j_{i\theta}(x-\epsilon) q_\eta(x)) \, . \tag{13}$$

Then the accumulation of the infinitesimal deformations gives rise to a flow of the deformed theories.

It is simple to see that commutativity of the charges, $[Q_i, Q_j] = 0$, along the flow, follows from (5) with (6) under the assumption that currents vanish at infinity: in this case, the flow is indeed a Hamiltonian flow generated by $X_{\theta\eta}$. But the asymptotic vanishing assumption requires inhomogeneous states, whereas here we are discussing homogeneous GGEs. Instead, it is possible to show the result in finite volumes, with periodic boundary conditions, and then infer the result at infinite volumes by taking the limit. In finite volumes, this is done directly using the transformation (12), without asymptotic assumptions.

For this purpose, assume that, at some point along the flow, the charges mutually commute.

Then the variation of the commutator $[Q_i, Q_j]$ along the flow at this point can be computed as

$$
\begin{aligned}
\delta[Q_i, Q_j] &= [\delta Q_i, Q_j] + [Q_i, \delta Q_j] \\
&= \int_{\mathbb{R}} \mathrm{d}\theta \int_{-\infty}^{\theta} \mathrm{d}\eta \, \delta\lambda_{\theta\eta} \\
&\quad \times \int_0^R \mathrm{d}x \left([q_\theta(x-\epsilon)j_{i\eta}(x), Q_j] - [j_{i\theta}(x-\epsilon)q_\eta(x), Q_j]\right) - \{i \leftrightarrow j\} \\
&= \int_{\mathbb{R}} \mathrm{d}\theta \int_{-\infty}^{\theta} \mathrm{d}\eta \, \delta\lambda_{\theta\eta} \\
&\quad \times \int_0^R \mathrm{d}x \left(q_\theta(x-\epsilon)[j_{i\eta}(x), Q_j] + [q_\theta(x-\epsilon), Q_j]j_{i\eta}(x)\right. \\
&\quad \left. - j_{i\theta}(x-\epsilon)[q_\eta(x), Q_j] - [j_{i\theta}(x-\epsilon), Q_j]q_\eta(x)\right) - \{i \leftrightarrow j\} \\
&= \mathrm{i} \int_{\mathbb{R}} \mathrm{d}\theta \int_{-\infty}^{\theta} \mathrm{d}\eta \, \delta\lambda_{\theta\eta} \int_0^R \mathrm{d}x \left(q_\theta(x-\epsilon)\partial_{t_j}j_{i\eta}(x) - \partial_x j_{j\theta}(x-\epsilon)j_{i\eta}(x)\right. \\
&\quad \left. + j_{i\theta}(x-\epsilon)\partial_x j_{j\eta}(x) - \partial_{t_j}j_{i\theta}(x-\epsilon)q_\eta(x)\right) - \{i \leftrightarrow j\} \, .
\end{aligned}
\tag{14}
$$

Using the generalised continuity equation $\partial_{t_i}j_{jk} = \partial_{t_j}j_{ik}$, which is valid up to $O(\delta\lambda_{\theta\phi})$ along the flow, we then find that

$$
\delta[Q_i, Q_j] = -\mathrm{i} \int_{\mathbb{R}} \mathrm{d}\theta \int_{-\infty}^{\theta} \mathrm{d}\eta \, \delta\lambda_{\theta\eta} \int_0^R \mathrm{d}x \, \partial_x \left(j_{j\theta}(x-\epsilon)j_{i\eta}(x) - j_{i\theta}(x-\epsilon)j_{j\eta}(x)\right) = 0 \, , \tag{15}
$$

where we invoked periodicity.

Therefore the charges are still in involution after the infinitesimal deformation, and thus remain so throughout the flow of the generalised $T\bar{T}$-deformation. Of course, the commutativity of the charges in a general basis implies that in the rapidity basis: $[Q_\theta, Q_\phi] = 0$ for any $\theta, \phi \in \mathbb{R}$

## 4.3 Flow equations for free energy and free energy fluxes

In models with commuting conserved charges $Q_\theta$, in addition to the usual free energy density $f$, there exists for each $\theta$ a free energy flux density $g_\theta$ [34]. Much like the free energy, these are generating functions of expectation values of current densities in GGEs. That is,

$$
\langle q_\theta \rangle = \frac{\delta f}{\delta\beta^\theta} \, , \quad \langle j_{\theta\phi} \rangle = \frac{\delta g_\theta}{\delta\beta^\phi} \, . \tag{16}
$$

By the commutativity shown above, $f$ and $g_\theta$ exist everywhere along the flow.

We find that the flow equations for the free energy density and fluxes are:

$$
\frac{\delta f}{\delta\lambda_{\theta\phi}} = g_\theta \frac{\delta f}{\delta\beta^\phi} - g_\phi \frac{\delta f}{\delta\beta^\theta} \, , \quad \frac{\delta g_\gamma}{\delta\lambda_{\theta\phi}} = g_\theta \frac{\delta g_\gamma}{\delta\beta^\phi} - g_\phi \frac{\delta g_\gamma}{\delta\beta^\theta} \, . \tag{17}
$$

Equations (17) fully specify the charge and current averages, and thus the thermodynamics, at arbitrary $\lambda_{\theta\phi}$, when solved with the initial condition that the thermodynamics be that of free particles at $\lambda_{\theta\phi} = 0$. Eqs. (17) are basis-independent. Written in a generic basis $\theta \to i$, $\phi \to j$, they are valid for deformations of any many-body system, whether integrable or not. These flow equations are one of the main results of this paper.

Eqs. (17) are derived again by putting the system on finite volume with periodic boundary conditions (and then taking the large-volume limit). This goes as follows.

The flow of $f$ can be obtained by invoking the Hellman-Feynman theorem applied to the conserved charges. In a GGE we have $\delta F/\delta\lambda_{\theta\phi} = \langle\delta W/\delta\lambda_{\theta\phi}\rangle$, where $F = Rf$ is the free energy and the conserved charge $W = \int_{\mathbb{R}} \mathrm{d}\theta\,\beta^{\theta}Q_{\theta}$ fixes the GGE weight. The resulting GGE average can be calculated by directly evaluating $\langle\delta Q_{\gamma}/\delta\lambda_{\theta\phi}\rangle = \langle\mathcal{O}_{\gamma\theta\phi}\rangle$ with (6). We use that $\langle\mathcal{O}_{\gamma\theta\phi}\rangle$ is independent of $\epsilon$, a well-known result which follows from the structure of the conservation laws and invariance of the state under space-time translations. Using clustering of the correlation function at large spatial separation, in the limit $\epsilon = R/2 \to \infty$, and finally the formula $g_{\theta} = -\int_{\mathbb{R}} \mathrm{d}\phi\,\beta^{\phi}\langle j_{\phi\theta}\rangle$ that stems from the Euler KMS relation [34], completes the proof.

For the free energy fluxes $g_{\theta}$, there is no Hellmann-Feynman theorem, and thus we must calculate the action of the deformation on both the currents which define the free energy flux density, and the state. This is possible to do, solely using general many-body principles, in particular the EKMS relation along with an important symmetry relation [5, 50–53] . Alternatively, if boost symmetry is present (e.g. translation invariant), the resulting self-conserved (generalised) currents can be used to derive the flow equations for $g_{\theta}$, under the assumption that the boost symmetry is preserved under the deformation. For example, in the $\delta$-Bose gas, the generalised currents $J_{i0} = \int_{\mathbb{R}} \mathrm{d}x \int_{\mathbb{R}} \mathrm{d}\theta\,\theta^{i} j_{\theta0}(x)$ of the particle number (i.e. $Q_0 = N$) are themselves conserved densities $J_{i0} = Q_{i-1}$.

With boost symmetry, in which case $\lambda_{\theta\phi} = \lambda_{\theta-\phi}$, the flow equations are also simplified,

$$\frac{\delta f}{\delta\lambda_{\theta}} = \int_{\mathbb{R}} \mathrm{d}\eta\,\langle q_{\eta}\rangle(g_{\eta+\theta} - g_{\eta-\theta}), \tag{18}$$

$$\frac{\delta g_i}{\delta\lambda_{\theta}} = \int_{\mathbb{R}} \mathrm{d}\eta\,\langle j_{i\eta}\rangle(g_{\eta+\theta} - g_{\eta-\theta}). \tag{19}$$

For brevity of the text, both proofs, with and without boost symmetry, are presented in Appendix C and D.

## 4.4 Comparison with the results of Hernández-Chifflet et al. (2020)

Having the flow equations for the generalised $T\bar{T}$-deformations at our disposal, it is worthwhile to recover those for another "generalised $T\bar{T}$-deformations" introduced in [16] for relativistic IQFTs. These deformations correspond to a particular choice of $\lambda_{\theta}$ in our deformations, which is the CDD factor $\lambda_{\theta} = \sum_{n:\mathrm{odd}} \alpha_n e^{-n\theta}$. The authors of [16] also derived the flow equations for the free energy fluxes in the infinite volume GGE, which, using our convention, are given by

$$\frac{\partial g_i}{\partial\alpha_n} = g_{-n}\frac{\partial g_i}{\partial\beta^n} - g_n\frac{\partial g_i}{\partial\beta^{-n}}. \tag{20}$$

Here, the basis defining the $\beta^n$ is chosen as $h_n(\theta) = e^{n\theta}$. Let us reproduce this from our boost invariant flow equation (19).

We have, from (19) and $\partial\lambda_{\theta}/\partial\alpha_n = e^{-n\theta}$,

$$\frac{\partial g_i}{\partial\alpha_n} = \int_{\mathbb{R}} \mathrm{d}\theta\,e^{-n\theta}\frac{\delta g_i}{\delta\lambda_{\theta}} = \int_{\mathbb{R}} \mathrm{d}\eta\,\langle j_{i\eta}\rangle(e^{n\eta}g_{-n} - e^{-n\eta}g_n) \tag{21}$$

where we used

$$g_n = \int_{\mathbb{R}} \mathrm{d}\theta\,h_n(\theta)g_{\theta}. \tag{22}$$

Further using

$$\int_{\mathbb{R}} \mathrm{d}\eta \, h_n(\eta) \langle j_{i\eta} \rangle = \langle j_{in} \rangle = \frac{\partial g_i}{\partial \beta^n} \tag{23}$$

we obtain the sought flow equation (20).

## 5 Thermodynamic Bethe ansatz

The thermodynamics of an integrable model is completely fixed by appropriate "scattering data": the one-particle eigenvalues $w(\theta)$ of the GGE weight $W$ (fixing the state), and $p(\theta)$ of the momentum $P$ (fixing the spatial direction), the particle statistics, and the S-matrix $S_\lambda(\theta, \phi)$. The average currents further require the extra ingredient of the dynamics, encoded within the one-particle eigenvalue $E(\theta)$ of the energy. Once these are known, the TBA formalism gives the free energy density and flux. The structure of TBA appears to be very universal [54–57]. Here we show that the generalised $T\bar{T}$-deformation provides a novel derivation of TBA. To be more precise, we shall show that by solving the flow equations of a generalised $T\bar{T}$-deformed theory parameterised by $\lambda_{\theta\phi}$, which was shown to be equivalent to the integrable model whose phase shift is $\lambda_{\theta\phi}$ in Sect. 3, we obtain the free energy and the free energy fluxes that coincide with those of the corresponding integrable system. This is achieved by using the explicit flow equations for $f$ and $g_\theta$ derived above, and showing that the TBA solves these equations with the appropriate free-particle initial condition (at $\lambda_{\theta\phi} = 0$). As the equations form a closed first-order system, a smooth solution is unique.

In general, again assuming a single particle species, the free energy density and fluxes are given, in TBA, by

$$f = -\int_{\mathbb{R}} \frac{\mathrm{d}p(\theta)}{2\pi} L(\varepsilon(\theta)), \quad g_i = -\int_{\mathbb{R}} \frac{\mathrm{d}h_i(\theta)}{2\pi} L(\varepsilon(\theta)), \tag{24}$$

where $\varepsilon$ is the pseudo-energy, solving a certain integral equation, and $-L(\varepsilon)$ is the free energy function, depending generally only on the particle statistics [33]. The expression for $g_i$ was derived for various classes of integrable models [44, 45, 58–61] and reads $2\pi g_\theta = -\mathrm{d}L(\varepsilon(\theta))/\mathrm{d}\theta$ in the asymptotic rapidity basis.

By (9), $f$ and $g_\theta$ depend on $\lambda_{\theta\phi}$ via the scattering phase $-\mathrm{i}\log S_\lambda(\theta, \phi) = -(\lambda_{\theta\phi} - \lambda_{\phi\theta})$. Elementary calculations show that they satisfy (17), see Appendix E. As the TBA manifestly gives the free-particle thermodynamics at constant $S(\phi, \theta)$ (the initial condition at $\lambda_{\theta\phi} = 0$), we have established, without the use of the Bethe ansatz or integrability structures, that the thermodynamics of generalised $T\bar{T}$-deformed free theories is that of integrable systems with corresponding scattering data. The particle statistics, momentum and other one-particle eigenvalues, and GGE, are encoded in the initial condition of the flow, while the S-matrix is generated by the deformation. This also establishes that generalised $T\bar{T}$-deformations of integrable systems preserve the TBA structure.

## 6 S-matrix Analyticity and Bound States

The family of deformations we have analysed is valid for any many-body system, relativistic or not, integrable or not. We have focussed on integrable systems, where the asymptotic states can be used for a basis of extensive charges, and obtained very general results. We now discuss in what sense the resulting S-matrix satisfies the properties expected for a well-defined theory.

As mentioned, the resulting S-matrix preserved unitarity because of the antisymmetry of the scattering phase. We note that the anti-symmetry of the kernel $\log S(\theta)$ in the

TBA formulation is in fact a fundamental constraint of many-body physics coming from the KMS relation [34]; this is thus satisfied by the generalised $T\bar{T}$-deformation. However, a few other requirements for a physical S-matrix are usually imposed. For simplicity of the following discussion, we focus on such requirements in relativistic QFT models. It is simple to guarantee that the (1+1)-dimensional Poincaré symmetry remains intact after deformation, with $\lambda_{\theta\phi} = \lambda_{\theta-\phi}$. We also assume that the initial condition of the flow is $S(\theta) = \pm 1$, the S-matrix for a free bosonic/fermionic system respectively.

First, as in [9], the generalised $T\bar{T}$-deformations do not need to realise UV complete QFTs. The deformed theory may have nontrivial small-distance physics, as is clear from [12,13,24]. Thus, strict locality may be broken, and this depends on the specific form of $\lambda_\theta$.

Second, general QFT considerations imply that the S-matrix can be analytically continued in rapidity space, with specific analyticity constraints related to the existence of bound states and other physical properties of the QFT. Without bound states, $S_\lambda(\theta)$ should be analytic for all $\theta$ in the physical strip $0 \leq \text{Im}(\theta) \leq \pi$ and should satisfy a number of properties, including unitarity and crossing symmetry. It turns out that such constraints restrict the form of $\lambda_\theta$ to a CDD factor: $\lambda_\theta - \lambda_{-\theta} = \sum_s \alpha_s \sinh(s\theta)$ for $s$ odd, the deformations considered in [9,14–16]. As recalled above, these values of $s$ are in fact the spins allowed for local charges, and this corresponds to the original Smirnov-Zamolodchimov deformations with local charges.

Yet, the deformation (9) holds for the physical S-matrix, with real rapidities, and imposes no such strong analyticity constraints: there is no reason to restrict $\lambda_\theta$ to CDD factors. In particular, deformations with quasi-local charges must break some analyticity requirements; although it is easy to preserve analyticity near to the real line may be preserved. A puzzle is that if $S_\lambda(\theta)$ has singularities which represent bound states according to the analytic S-matrix theory, the nuclear democracy principle states that new particle species must be present in the spectrum. Where are these particles in the deformed S-matrix and TBA?

Although we do not have the full answer, a likely scenario is as follows. In such cases, Eq. (9) only gives the scattering amplitudes *for the deformation of the original particle species*. That is, there are particle species in the deformed theory in one-to-one correspondence with those of the original particles. There may be other asymptotic particles, for instance involving stable compounds, but we have not obtained their amplitudes. These must be evaluated from (9) by bootstrap methods [2]. Thus the deformed theory S-matrix, in such cases, only describe *a part of the full spectrum of asymptotic particles*.

What about the thermodynamics and the TBA equations? The TBA equations we have obtained only involve the original particle species. Here, the principle is the fact that asymptotic particle densities are in one-to-one correspondence with extensive conserved quantities. With more asymptotic particle species, such as bound states, new conserved quantities emerge, but in general, it is perfectly allowed to consider GGEs that only involve the subset of the extensive conserved quantities corresponding to the original particle species. One only needs to bring to zero the "temperatures" associated to the new asymptotic particles. This is what happens in the TBA formulation we have obtained. It is also important to note that the submanifold of the full GGE of the deformed model that is spanned by the original particle is also invariant under hydrodynamic evolution (much like gases of KdV solitons without phonon modes [62] and zero-entropy GHD [63]), because the asymptotic particle densities are not only extensive conserved quantities, but also hydrodynamic normal modes.

It would be interesting to analyse more precisely a specific model with new bound states appearing under deformation, and construct the full scattering matrix. We emphasise that, of course, if no new particle species appears, our results give the full deformed S-matrix, and the TBA for the full deformed GGE.

Finally, with this partial knowledge of the S-matrix, it is not possible to fully address many of the other properties expected of QFT, such as PCT and crossing symmetry. Indeed, PCT and

crossing symmetry require the knowledge of charge-conjugated particles, which may require the analysis of the analytic structure and the determination of new particles in the asymptotic spectrum.

The analysis can be extended to situations where the initial Hamiltonian contains several species of particle; $\theta$ becomes a double index, gathering the rapidity and the particle species. The generalisation of the deformation to this situation is presented in Appendix B.2. It is able to produce arbitrary diagonal scattering matrices, as well as certain types of non-diagonal scattering. In all cases, we believe that a well-defined many-body system is obtained after deformation without strong constraints on the analytic structure of $\lambda_\theta$.

# 7 Conclusion

In this paper we explored the space of integrable systems by introducing a new class of integrable deformations of Smirnov-Zamolodchikov type. They are based on the scattering basis for the complete space of extensive conserved charges, and contain the deformations considered by Smirnov and Zamolodchikov as a particular subset. Concentrating on theories with a single particle species, we showed that these generalised $T\bar{T}$-deformations can yield generic S-matrices. We showed from basic principles, without using integrability, that the thermodynamics of the deformed theories coincides with that obtained by TBA. In particular, this gives an alternative way of deriving the GGE averages of currents, which does not involve the Bethe ansatz [5, 46, 47] or the assumption of boost invariance [47]. Combining the $\eta$-deformations of the energy, the boost $\kappa$-deformations, and the generalised $T\bar{T}$-deformations, one obtains generic deformations of all main ingredients of TBA, thus generically, we conjecture, the full tangent space $T\Sigma^{\mathrm{Int}}$. This can generate a large class of integrable models, quantum or classical.

When the deformed theory admits bound states of the original particles, the S-matrix (9) must be supplemented by matrix elements involving the bound states (obtained by bootstrap), and the flow equations (17) only reproduce a (stable) subset of the GGE of the deformed theory. At points of the deformation where bound states appear, the manifold $\Sigma^{\mathrm{Int}}$ is singular. At such points, reconstructing the full GGE within the formalism of generalised $T\bar{T}$ deformations is an important open question.

If the deformed theory happens to have the S-matrix of a known model, these theories are equivalent by a general principle of IQFT. The S-matrix fixes the set of all form factors (matrix elements of local operators in asymptotic states), by the bootstrap principle. Hence it fixes the set of local operators and their correlations. The precise relation between the local operators, as expressed in terms of fields, is however highly nontrivial. It is likely similar to that between the sine-Gordon model and the Thirring model: it has long been known that, while the local observable structures of the two models differ, the two models possess the same S-matrix, and are hence equivalent [64].

Finally, although we implicitly focused on the massive case, generalised $T\bar{T}$-deformations are equally applicable to massless models (with a slight abuse of the notion of asymptotic states). In particular, one can deform a CFT without introducing scattering between right and left movers, which should yield arbitrary CFTs according to [65]. It would be very interesting to work out specific examples.

## Acknowledgements

BD is grateful to O. A. Castro-Alvaredo for comments and discussions. TY thanks M. Medenjak for comments. JD acknowledges funding from the EPSRC Centre for Doctoral Training in Cross-Disciplinary Approaches to Non-Equilibrium Systems (CANES) under grant EP/L015854/1.

## A Quasi-local charges in free Majorana fermions

It is instructive to see how quasi-local charges can be obtained in free systems, such as the free Majorana fermion theory, whose Hamiltonian reads

$$H = \frac{1}{2} \int_0^R dx \left[ i\nu(\psi_R \partial_x \psi_R - \psi_L \partial_x \psi_L) + 2i\psi_R \psi_L \right], \qquad (25)$$

where $\psi_R$ and $\psi_L$ are real chiral fermions, $\nu$ is the Fermi velocity (in the interpretation of the Majorana fermion as the emergent low-energy mode near the Fermi edge), and we set $m = 1$. The Hamiltonian is diagonalised in the infinite volume as

$$H = \int_{\mathbb{R}} d\theta \, \cosh(\theta) Q_\theta, \quad Q_\theta := Z_\theta^\dagger Z_\theta, \qquad (26)$$

where $Z_\theta$ satisfies $\{Z_\theta^\dagger, Z_{\theta'}\} = \delta(\theta - \theta')$, and is given by a linear combination of $\psi_R$ and $\psi_L$. Clearly any charge of the form $W = \int_{\mathbb{R}} d\theta \, \beta^\theta Q_\theta$ is conserved, and a GGE is thus characterised by a potential function $\beta^\theta$ of $\theta$. The only requirement is that it should guarantee clustering of correlation functions; for instance, a sufficient condition is that $\beta^\theta$ be analytic in neighbourhood of $\mathbb{R}$ and sufficiently increasing as $|\theta| \to \infty$. More related to the present work, at the linearised level, describing the tangent to the manifold of states, $\delta\beta^\theta$ constitute a (un-normalisable) basis for the space $L^2(\mathbb{R}, \rho_F)$ with $\rho_F(\theta) = (1 + e^{\beta^\theta})^{-1}$ the Fermi density of states [31].

The charge can also be written as

$$W = \int_{\mathbb{R}^2} dx dy \, \psi^\dagger(x) \tilde{\beta}(x - y) \psi(y), \qquad (27)$$

where $\tilde{\beta}(x) = \int_{\mathbb{R}} d\theta \, \beta^\theta e^{-ip(\theta)x}$ with momentum $p(\theta) = \sinh(\theta)$, and $\psi$ is some linear combination of $\psi_R$ and $\psi_L$. Quasi-locality of the charge is ensured by requiring that $\tilde{\beta}(x)$ decay exponentially. Such a choice of $\beta^\theta$ is always possible, and one example was found in [41].

## B Deformed 2-Particle S-matrix

We provide the calculations of the 2-body scattering shifts induced by the $\lambda$-deformations. The calculations are performed for classical particle systems, where they are simplest; we follow [13]. For quantum systems, the required proof is a simple generalisation of that presented in full detail in Ref. [12], section 3.2 and Appendix B, and is thus not repeated. The trivial alteration required is taking the sum over all $\nu$ with the terms $h_\nu(\theta) = \delta(\theta - \nu)$, and the calculation follows immediately.

## B.1 Single Particle Species

In order to solve the scattering problem in the deformed system, we consider a system of particles on the infinite line. As shown in the main text, throughout the flow the system admits infinitely many conserved charges, hence it stays integrable. The presence of infinitely many conserved charges guarantees, by standard arguments, scattering is elastic and factorised (the argument from Parke was made in QFT [49], but holds more generally, for instance in classical particle systems as emphasised in [33]).

We assume that there is originally a single species of particle. This means that it is sufficient to restrict to this particle species on the full flow. As discussed in the main text, the $\lambda$-deformation may introduce attractive interactions and bound states, which would then form new asymptotic states. However, it is consistent to consider the original particle species. Indeed, this species still forms part of the asymptotic spectrum, and, as is shown in the calculation below, this particle keeps its scattering diagonal, hence in scattering events, it does not mix with other particle types that may be admitted under the deformation.

As we are interested in analysing scattering, it is sufficient to describe the trajectories, and their deformations, in the asymptotic in- and out-regions $t \to \mp\infty$. In these regions, particles are well separated from each other and keep their ordering over time. The calculation is simplest with the choice of labelling in which the position ordering agrees with the label ordering: $x_n(t) > x_m(t)$ if $n > m$; this labelling is assumed for both $t \to \mp\infty$. Note that if particles have a hard core (with potentially additional interactions), the labelling simply follows the particle, as particles don't go through each other. On the other hand if particles can go through each other, then any given label does not follow any given particle (see a discussion about the relation between the "hard-core" and "go-through" pictures in [13]). Here and below, $x_n, p_n$ are the canonical coordinates.

Using (7) and our labelling assumption, we then have that, in both regions $t \to \mp\infty$,

$$X_{\theta\phi} = -\sum_{n>m} \delta(\theta - p_m)\delta(\phi - p_n). \tag{28}$$

We can use this result to calculate the flow of the canonical variables with respect to $\lambda_{\theta\phi}$, where we have trivially that $\{p_i, X_{\theta\phi}\} = 0$, and (recall that we restrict to $\theta \geq \phi$):

$$\{x_i, X_{\theta\phi}\} = \partial_\theta \sum_{n>i} \delta(\theta - p_i)\delta(\phi - p_n) + \partial_\phi \sum_{i>n} \delta(\theta - p_n)\delta(\phi - p_i), \tag{29}$$

where we shifted the derivatives onto the variables $\theta, \phi$.

We would like to construct the solution for all $\lambda_{\theta\phi} \neq 0$. As we are in the asymptotic in- and out-regions, at large $t \to \mp\infty$, we have $x_i(t) \sim p_i^{\text{in,out}} t + x_i^{\text{in,out}}$, where $p_i^{\text{in,out}}$ are the asymptotic momenta and $x_i^{\text{in,out}}$ are the impact parameters. Therefore, positions are widely separated, and finite deformations along the flow cannot change their order. It is then a simple matter to solve the above equations. We obtain the following momentum shifts due to the deformation:

$$\Delta p_i = 0, \tag{30}$$

and the following position shifts (in the asymptotic regions),

$$\Delta x_i = -\sum_{i>n} \Theta(p_n - p_i)\partial_{p_i}\lambda_{p_n p_i} - \sum_{n>i} \Theta(p_i - p_n)\partial_{p_i}\lambda_{p_i p_n}, \tag{31}$$

where $\Theta(p)$ is the step function. As is clear from (28) with $\theta > \phi$, in the asymptotic in-region the deformation is nontrivial as $p_i^{\text{in}} > p_{i+1}^{\text{in}}$, while in the asymptotic out region, it is trivial as momenta have exchanged their order.

The S-matrix is calculated in the usual fashion by solving the 2-particle problem. We are looking for the scattering shifts of the "quasi-particles" of the resulting integrable model: these are the momentum tracers. Using, in the asymptotic regions, $p_1 \equiv p_1^{\text{in}} > p_2 \equiv p_2^{\text{in}}$ and $p_2^{\text{out}} > p_1^{\text{out}}$, the scattering shifts $\varphi_{1,2}$ of the momentum tracers for $p_1$ and $p_2$ (by convention, the amount shifted to the left (right) for $p_1$ ($p_2$)), are

$$\Delta x_1^{\text{in}} - \Delta x_2^{\text{out}} = \varphi_1 = -\partial_{p_1} \lambda_{p_1 p_2}, \tag{32}$$

$$\Delta x_1^{\text{out}} - \Delta x_2^{\text{in}} = \varphi_2 = \partial_{p_2} \lambda_{p_1 p_2}. \tag{33}$$

Using $\varphi_1 = \partial_{p_1}(-\mathrm{i} \log S(p_1, p_2))$ and $\varphi_2 = \partial_{p_2}(-\mathrm{i} \log S(p_2, p_1))$, this leads us to identify:

$$S_\lambda(p, q) = e^{-\mathrm{i}(\lambda_{pq} - \lambda_{qp})} S(p, q). \tag{34}$$

This construction gives a Hamiltonian version of the "flea gas" algorithm [25]. The latter takes care, in an algorithmically sensible way but not as a Hamiltonian time evolution, of the potential particle ordering problems that may emerge at finite densities when introducing momentum-dependent scattering lengths. It would be interesting to analyse the precise effect of the $\lambda$ flow discussed here not just in the asymptotic large-time regions but at finite times, in order to see how this Hamiltonian formulation takes care of such ordering issues.

## B.2 Multiple Particle Species

We now extend the above construction to a system with several particle species, labelled by the Latin index $i = 1, \cdots, N$. We assume that the undeformed model has diagonal scattering amongst species; again as we will see, this stays true under deformation, hence it is still sufficient to consider the sector of scattering involving only these species, even if, under deformation, more species may appear in the asymptotic spectrum.

For this situation, it is more natural to keep together species with momenta. In this description, particle order is changed between the in- and out-regions. It is convenient to still use the canonical variables $x_n, p_n$, and associated a species value $i_n$ to each pair. We will assume the ordering $x_1 < x_2 < \ldots < x_N$ as $t \to \infty$, and hence the opposite order as $t \to \infty$. We then have $p_1 > p_2 > \ldots > p_N$ at both $t \to \pm\infty$.

We can then introduce the generator of the deformation:

$$X_{\theta\phi}^{ij} = -\int_0^R dy \int_0^{y-\epsilon} dx \, q_\theta^i(x) q_\phi^j(y), \quad \theta \geq \phi, \tag{35}$$

where the charge densities now carry a dual label, referring to the rapidity and species. Again we can write an explicit expression for the charge density, which is valid throughout the flow in the in- and out-regions:

$$q_\theta^i(x) = \sum_{n=1}^N \delta(x - x_n) \delta(\theta - \theta(p_n)) \delta_{i,i_n}. \tag{36}$$

Taking into account the particle orderings in the in and out asymptotic regions, we then have

$$X_{\theta\phi}^{ij} = \begin{cases} -\sum_{n>m} \delta(\theta - p_m) \delta(\phi - p_n) \delta_{i,i_m} \delta_{j,i_n} & (t \to -\infty), \\ -\sum_{n<m} \delta(\theta - p_m) \delta(\phi - p_n) \delta_{i,i_m} \delta_{j,i_n} & (t \to \infty). \end{cases} \tag{37}$$

As in the single-species case (which was done with a different, "hard-core" labelling), $X_{\theta\phi}^{ij}$ is nontrivial (trivial) in the in (out) region, as follows from the ordering on momenta. A calculation as in section B.1 gives the following deformed scattering matrix:

$$S_\lambda(p, q)_{ij} = e^{-\mathrm{i}(\lambda_{pq}^{ij} - \lambda_{qp}^{ji})} S(p, q)_{ij}, \tag{38}$$

where $S(p,q)_{ij}$ is the two-body scattering amplitude of particles of momentum $p$ and species $i$, against particle of momentum $q$ and species $j$, in the un-deformed model. This reproduces arbitrary diagonal scattering, where two-body processes represent pure transmission, with zero reflection amplitudes.

We finally mention that it is also possible to consider non-diagonal scattering, with reflection and transmission. We give here some pointers, although this is not used in the present paper. We represent the non-diagonal scattering in the classical realm, by assuming that particle types reflect or transmit with probabilities $|A_R^{ij}|^2$ and $|A_T^{ij}|^2$, respectively, and for simplicity without any phase shift. This represents classically the quantum scattering

$$S_T^{ij} = |A_T^{ij}|, \ \ S_R^{ij} = |A_R^{ij}|, \ \ i \neq j. \tag{39}$$

We perform the $\lambda$ deformation, which can be obtained again from the explicit expression for the charge density (36). The $\lambda$-deformation does not change the probabilities of reflection and transmission, but introduces shifts, which (we assume) induce phases to the quantum scattering amplitude in the usual way,

$$S_T^{ij} = |A_T^{ij}|e^{i\delta_T^{ij}}, \ \ S_R^{ij} = |A_R^{ij}|e^{i\delta_R^{ij}}, \ \ i \neq j. \tag{40}$$

Solving the classical problem, we find the following scattering phases:

$$\delta_R^{ij}(\theta, \phi) = -\left(\lambda_{\theta\phi}^{ij} - \lambda_{\phi\theta}^{ij}\right), \ \ \delta_T^{ij}(\theta, \phi) = -\left(\lambda_{\theta\phi}^{ij} - \lambda_{\phi\theta}^{ji}\right). \tag{41}$$

Unitarity imposes that $|A_T|^2 + |A_R|^2 = 1$, and also the non-trivial condition:

$$\lambda_{\theta\phi}^{ij} - \lambda_{\theta\phi}^{ji} = \left(n + \frac{1}{2}\right)\pi, \ \ n \in \mathbb{Z}. \tag{42}$$

It would be interesting to further study this construction, in order to establish the space of reflective two-body processes that this deformation gives access to.

## C Flow Equations for Free Energy Fluxes: Generic Derivation

In this section we will derive the flow equation for the free energy fluxes under the generalised $T\bar{T}$-deformation. The results of this section are basis-independent: they are independent of the chosen index set for the charges. For instance, taking the indices to be rapidities in the final result (64) returns (17) in the main text. The results are based solely on general principles of many-body physics and statistical mechanics, and do not involve concepts of integrability or QFT.

### C.1 Action of the $X$ generator

In this section we calculate the effect of the deformation generator $i[X_{ab}, \cdot]$ ($X_{ab}$ is from Eq. (7)) on the charge and current densities. We consider a homogeneous thermodynamic state in a system of length $R$ with periodic boundary conditions. We take $R$ large enough, which will be taken to infinity later on. We consider a small but nonzero $\iota > 0$, and positions $\iota < x < R - \iota$; this avoids additional boundary terms and is sufficient for this derivation.

Using that $i[Q_i, j_{kj}] = \partial_{t_i} j_{kj}$, we deduce, for all $\iota < x < R - \iota$,

$$i \int_0^y dz \, [q_i(z), j_{kj}(x)] = \partial_{t_i} j_{kj}(x) \chi(x < y) + \mathcal{O}_{ikj}(x, y), \tag{43}$$

where $\mathcal{O}_{ikj}(x, y)$ is nonzero only around $x = y$, and is a local observable supported at $x$ for $x \simeq y$. $\chi(x < y)$ is the indicator function that gives 1 if $x < y$ and gives zero otherwise. Likewise we have

$$\mathrm{i} \int_z^R \mathrm{d}y\, [q_i(y), j_{kj}(x)] = \partial_{t_i} j_{kj}(x) \chi(x > z) - \mathcal{O}_{ikj}(x, z). \tag{44}$$

We can now evaluate the effect of the deformation on the currents (here we take the point-splitting parameter $\epsilon = 0$ for simplicity):

$$
\begin{aligned}
-[\mathrm{i}X_{ab}, j_{ki}(x)] &= \int_0^R \mathrm{d}y \left( \partial_{t_a} j_{ki}(x) \chi(x < y) q_b(y) + \mathcal{O}_{aki}(x, y) q_b(y) \right) \\
&+ \int_0^R \mathrm{d}z \left( q_a(z) \partial_{t_b} j_{ki}(x) \chi(x > z) - q_a(z) \mathcal{O}_{bki}(x, z) \right) \\
&= \partial_{t_k} j_{ai}(x) \int_x^R \mathrm{d}y\, q_b(y) + \int_0^x \mathrm{d}z\, q_a(z) \partial_{t_k} j_{bi}(x) + A_{abki}(x), \quad (45)
\end{aligned}
$$

where

$$A_{abki}(x) = \int_0^R \mathrm{d}y\, \mathcal{O}_{aki}(x, y) q_b(y) - \int_0^R \mathrm{d}z\, q_a(z) \mathcal{O}_{bki}(x, z), \tag{46}$$

is a local observable at $x$. Therefore

$$
\begin{aligned}
[\mathrm{i}X_{ab}, j_{ki}(x)] &= -\partial_{t_k}\left( j_{ai}(x) \int_x^R \mathrm{d}y\, q_b(y) + \int_0^x \mathrm{d}z\, q_a(z) j_{bi}(x) \right) \\
&+ j_{ai}(x) j_{kb}(x) - j_{ai}(x) j_{kb}(R) - j_{ka}(x) j_{bi}(x) + j_{ka}(0) j_{bi}(x) - A_{abki}(x).
\end{aligned}
\tag{47}
$$

By noticing that the total momentum $P = p^k Q_k$ for some set $p^k$, we see that $q_i = p^k j_{ki}$, and thus by contracting (47) with $p^k$ we obtain the deformation of charge densities

$$
\begin{aligned}
[\mathrm{i}X_{ab}, q_i(x)] &= \partial\left( j_{ai}(x) \int_x^R \mathrm{d}y\, q_b(y) + \int_0^x \mathrm{d}z\, q_a(z) j_{bi}(x) \right) \\
&+ j_{ai}(x) q_b(x) - j_{ai}(x) q_b(R) - q_a(x) j_{bi}(x) + q_a(0) j_{bi}(x) - p^k A_{abki}(x).
\end{aligned}
\tag{48}
$$

Here $\partial = -p^k \partial_{t_k} = -\mathrm{i}[P, \cdot]$ is the space-derivative operator. Note how it is different from the derivative with respect to the parameter $x$. Replacing $\partial$ by $\partial_x$ can be achieved by simultaneously deleting the terms $+j_{ai}(x) q_b(R)$ and $-q_a(0) j_{bi}(x)$ in the second line, because of the boundary terms at $R$ and $0$ in the first line. We can then use the methods of Refs. [34, 43] to establish

$$p^k A_{abki} = (j_{ai}(x) + j_{ia}(x)) q_b(x) - q_a(x)(j_{bi}(x) + j_{ib}(x)), \tag{49}$$

and we obtain

$$
\begin{aligned}
[\mathrm{i}X_{ab}, q_i(x)] &= \partial\left( j_{ai}(x) \int_x^R \mathrm{d}y\, q_b(y) + \int_0^x \mathrm{d}z\, q_a(z) j_{bi}(x) \right) \\
&- j_{ia}(x) q_b(x) - j_{ai}(x) q_b(R) + q_a(x) j_{ib}(x) + q_a(0) j_{bi}(x).
\end{aligned}
\tag{50}
$$

Note that we cannot obtain the commutator $\mathrm{i}[X_{a,b}, Q_i]$ by integrating (50) over $x \in [0, R]$: Eq. (50) is valid only for $\iota < x < R_\iota$. For $x$ near to $0 \equiv R$, additional boundary terms appear. Taking into account such boundary terms, the commutator with the total charge is instead given by Eqs. (5), (6) and (8), that is:

$$\frac{\partial Q_i}{\partial \lambda_{ab}} = \mathrm{i}[X_{ab}, Q_i] + Q_a j_{ib}(0) - j_{ia}(0) Q_b. \tag{51}$$

## C.2 Deformations of densities and currents

We now define the deformations of charge densities and currents as (for the charge density, this is in agreement with Eqs. (5), (6))

$$\frac{\partial q_i}{\partial \lambda_{ab}} = q_a j_{ib} - j_{ia} q_b, \quad \frac{\partial j_{ki}}{\partial \lambda_{ab}} = j_{ai} j_{kb} - j_{ka} j_{bi} - A_{abki}. \tag{52}$$

With these definitions, we have

$$\frac{\partial q_i}{\partial \lambda_{ab}} = [iX_{ab}, q_i] + j_{ai}(x) q_b(R) - q_a(0) j_{bi}(x) - \partial o_{abi} \tag{53}$$

$$\frac{\partial j_{ki}}{\partial \lambda_{ab}} = [iX_{ab}, j_{ki}] + j_{ai}(x) j_{kb}(R) - j_{ka}(0) j_{bi}(x) + \partial_{t_k} o_{abi}, \tag{54}$$

where

$$o_{abi}(x) = j_{ai}(x) \int_x^R dy \, q_b(y) + \int_0^x dz \, q_a(z) j_{bi}(x). \tag{55}$$

This implies that the conservation laws are preserved "in the bulk": with $\iota < x < R - \iota$, using the fact that both $\partial/\partial \lambda_{ab}$ and $[iX_{ab}, \cdot]$ are derivations, and using (51), we obtain

$$\frac{\partial}{\partial \lambda_{ab}} (\partial_{t_k} q_i + \partial j_{ki}) = \frac{\partial}{\partial \lambda_{ab}} (\partial_{t_k} j_{li} - \partial_{t_l} j_{ki}) = 0. \tag{56}$$

## C.3 Deformation of free energy fluxes

Now we need to calculate the action of the deformation on the expectation values of currents $\langle j_{ki} \rangle$ in GGE states parameterised by a density matrix $\rho \propto \exp(-W)$, where $W = \beta^i Q_i$. We require that the state be clustering, that is $\langle \mathcal{O}_1(x) \mathcal{O}_2(y) \rangle \to \langle \mathcal{O}_1 \rangle \langle \mathcal{O}_2 \rangle$ for $\mathrm{dist}(x, y) \propto R$. Using (54), we then have

$$\langle [iX_{ab}, j_{ki}(x)] \rangle = \langle \partial_{\lambda_{ab}} j_{ki} \rangle - \langle j_{ai} \rangle \langle j_{kb} \rangle + \langle j_{ka} \rangle \langle j_{bi} \rangle. \tag{57}$$

We now evaluate the left-hand side using cyclicity of the trace

$$\begin{aligned}
\langle [iX_{ab}, j_{ki}] \rangle &= Z^{-1} \mathrm{Tr}\left( e^{-W} [iX_{ab}, j_{ki}] \right) \\
&= -Z^{-1} \mathrm{Tr}\left( [iX_{ab}, e^{-W}] j_{ki} \right) \\
&= Z^{-1} \int_0^1 du \, \mathrm{Tr}\left( e^{-uW} [iX_{ab}, W] e^{(u-1)W} j_{ki} \right).
\end{aligned} \tag{58}$$

We use the transformation equation (8) in order to evaluate $[iX_{ab}, W]$:

$$[iX_{ab}, W] = \frac{\partial W}{\partial \lambda_{ab}} - \beta^\ell [Q_a j_{\ell b}(0) - j_{\ell a}(0) Q_b]. \tag{59}$$

We define $\mathcal{O}^u = e^{-uW} \mathcal{O} e^{uW}$, and notice that this operator is still local and therefore clustering holds for all $u$ (locality under imaginary time evolution can be shown in quantum spin chains). Then, omitting $-\beta^\ell \int_0^1 du$ and using clustering, the expectation values of the last two terms of (59) as included in (58) are

$$\begin{aligned}
&\langle Q_a j_{\ell b}^u(0) j_{ki}(x) - j_{\ell a}^u(0) Q_b j_{ki}(x) \rangle \\
&= \langle Q_a j_{\ell b}^u(0) j_{ki}(x) - j_{\ell a}^u(0) Q_b j_{ki}(x) \rangle^c + \langle Q_a \rangle \langle j_{\ell b} \rangle \langle j_{ki} \rangle - \langle Q_b \rangle \langle j_{\ell a} \rangle \langle j_{ki} \rangle \\
&= \left( -\frac{\partial}{\partial \beta^a} + \langle Q_a \rangle \right) [\langle j_{\ell b} \rangle \langle j_{ki} \rangle] - \left( -\frac{\partial}{\partial \beta^b} + \langle Q_b \rangle \right) [\langle j_{\ell a} \rangle \langle j_{ki} \rangle].
\end{aligned} \tag{60}$$

We now use (57), then (58) with (59) and (60), as well as the EKMS relation [34]

$$\beta^\ell j_{\ell k} = -g_k \tag{61}$$

and the symmetry relation [5, 50–53]

$$\frac{\partial \langle j_{ka} \rangle}{\partial \beta^b} = \frac{\partial \langle j_{kb} \rangle}{\partial \beta^a}, \tag{62}$$

and obtain:

$$
\begin{aligned}
\frac{\partial \langle j_{ki} \rangle}{\partial \lambda_{ab}} &= -\int_0^1 du \, \langle \left( \frac{\partial W}{\partial \lambda_{ab}} \right)^u j_{ki} \rangle^c + \langle \frac{\partial j_{ki}}{\partial \lambda_{ab}} \rangle \\
&= -\int_0^1 du \, \langle \left( \frac{\partial W}{\partial \lambda_{ab}} \right)^u j_{ki} \rangle^c + \langle [iX, j_{ki}] \rangle + \langle j_{ai} \rangle \langle j_{kb} \rangle - \langle j_{ka} \rangle \langle j_{bi} \rangle \\
&= \langle \partial_{\lambda_{ab}} W \rangle \langle j_{ki} \rangle - \beta^\ell \left( -\frac{\partial}{\partial \beta^a} + \langle Q_a \rangle \right) \left[ \langle j_{\ell b} \rangle \langle j_{ki} \rangle \right] \\
&\quad + \beta^\ell \left( -\frac{\partial}{\partial \beta^b} + \langle Q_b \rangle \right) \left[ \langle j_{\ell a} \rangle \langle j_{ki} \rangle \right] + \langle j_{ai} \rangle \langle j_{kb} \rangle - \langle j_{ka} \rangle \langle j_{bi} \rangle \\
&= \beta^\ell \frac{\partial}{\partial \beta^a} \left[ \langle j_{\ell b} \rangle \langle j_{ki} \rangle \right] - \beta^\ell \frac{\partial}{\partial \beta^b} \left[ \langle j_{\ell a} \rangle \langle j_{ki} \rangle \right] + \langle j_{ai} \rangle \langle j_{kb} \rangle - \langle j_{ka} \rangle \langle j_{bi} \rangle \\
&= \beta^\ell \langle j_{\ell b} \rangle \frac{\partial}{\partial \beta^a} \langle j_{ki} \rangle - \beta^\ell \langle j_{\ell a} \rangle \frac{\partial}{\partial \beta^b} \langle j_{ki} \rangle + \langle j_{ai} \rangle \langle j_{kb} \rangle - \langle j_{ka} \rangle \langle j_{bi} \rangle \\
&= -g_b \frac{\partial}{\partial \beta^i} \langle j_{ka} \rangle + g_a \frac{\partial}{\partial \beta^i} \langle j_{kb} \rangle + \langle j_{ai} \rangle \langle j_{kb} \rangle - \langle j_{ka} \rangle \langle j_{bi} \rangle \\
&= -\frac{\partial}{\partial \beta^i} (g_b \langle j_{ka} \rangle) + \frac{\partial}{\partial \beta^i} (g_a \langle j_{kb} \rangle). \tag{63}
\end{aligned}
$$

Using finally that $\langle j_{ki} \rangle = \partial_{\beta^i} g_k$, we obtain the desired result:

$$\frac{\partial g_k}{\partial \lambda_{ab}} = g_a \langle j_{kb} \rangle - g_b \langle j_{ka} \rangle. \tag{64}$$

This shows Eq. (17) in the main text.

## D  Flow Equations from a Self-Conserved Current

It is also possible to show the free energy flux flow equations from the existence of a self-conserved current, or bridging pair. For example, in a free Galilean system, conserved charges and generalised currents can be constructed from the particle density operator $Q_\theta$ according to

$$J_{ij} = \int_{\mathbb{R}} d\theta \, h_j(\theta) Q_\theta h_i'(\theta), \tag{65}$$

from which it immediately follows that $J_{i0} = Q_{i-1}$ and $J_{ij} = J_{j+1,i-1}$, where $Q_0 = N$, the particle number. In fact the existence of boost symmetry predicts the existence of a bridging pair [61]. Therefore, we assume that the bridging pair stays intact provided that the system is deformed without breaking boost symmetry. This is accomplished by having the deformation parameter explicitly boost invariant $\lambda_{\theta,\phi} = \lambda_{\theta-\phi}$. In this case the charges in the physical basis are deformed as

$$\delta Q_i = \int_{\mathbb{R}} d\theta \int_{-\infty}^{\theta-\epsilon} d\eta \, \delta\lambda_{\theta-\eta} \mathcal{O}_{i\theta\eta} = \int_0^\infty d\theta \delta\lambda_\theta \int_{\mathbb{R}} d\eta \, \mathcal{O}_{i,\theta+\eta,\eta}. \tag{66}$$

To make the exposition below concrete, we consider the Lieb-Liniger $\delta$-Bose gas, where the bridging pair is again $J_{i0} = Q_{i-1}$ (physical charges are labelled as $Q_0 = N, Q_1 = P, Q_2 = H$). Moreover we will work with charges in the physical basis, as the bridging pair connects a charge and a current in that basis. We will make use of the bridging pair to derive the flow equations for all the free energy fluxes, using the flow equation for the free energy, which can be generally derived using only the Hellmann-Feynman theorem. Specifically, as a result of the bridging pair, the following relation holds

$$\frac{\partial}{\partial \beta^{i-1}} \frac{\delta f}{\delta \lambda_\theta} = \frac{\partial}{\partial \beta^0} \frac{\delta g_i}{\delta \lambda_\theta}. \tag{67}$$

This is the crucial identity in the following proof. As in the proof of commutativity of charges, we assume that the boost symmetry is preserved by the deformation, and consequently that $J_{i0} = Q_{i-1}$ and $J_{ij} = J_{j+1,i-1}$ remain valid. The bridging pair implies the identity $\partial_{\beta^k} \langle q_{i-1} \rangle = \partial_{\beta^{i-1}} \langle q_k \rangle = \partial_{\beta^k} \langle j_{i0} \rangle = \partial_{\beta^0} \langle j_{ik} \rangle$, i.e.

$$\int_{\mathbb{R}} \mathrm{d}\theta \, h_k(\theta) \frac{\partial}{\partial \beta^0} \langle j_{i,\theta} \rangle = \int_{\mathbb{R}} \mathrm{d}\theta \, h_k(\theta) \frac{\partial}{\partial \beta^{i-1}} \langle q_\theta \rangle, \tag{68}$$

from which we get

$$\frac{\partial}{\partial \beta^0} \langle j_{i,\theta} \rangle = \frac{\partial}{\partial \beta^{i-1}} \langle q_\theta \rangle. \tag{69}$$

Another useful identity is obtained from $\langle j_{i0} \rangle = \langle q_{i-1} \rangle$. Namely

$$\int_{\mathbb{R}} \mathrm{d}\theta \, h_i(\theta) \frac{\partial}{\partial \beta^0} g_\theta = \int_{\mathbb{R}} \mathrm{d}\theta \, h_{i-1}(\theta) \langle q_\theta \rangle = -\int_{\mathbb{R}} \mathrm{d}\theta \, h_i(\theta) \partial_\theta \langle q_\theta \rangle, \tag{70}$$

which gives

$$\frac{\partial}{\partial \beta^0} g_\theta = -\partial_\theta \langle q_\theta \rangle. \tag{71}$$

The final required identity follows by writing $\langle j_{ij} \rangle$ in two ways. First we have

$$\langle j_{ij} \rangle = \int_{\mathbb{R}} \mathrm{d}\theta \, h_j(\theta) \langle j_{i\theta} \rangle = -\int_{\mathbb{R}} \mathrm{d}\theta \, h_{j+1}(\theta) \partial_\theta \langle j_{i\theta} \rangle. \tag{72}$$

Equating this with the alternative expression

$$\langle j_{ij} \rangle = \langle j_{j+1,i-1} \rangle = \int_{\mathbb{R}} \mathrm{d}\theta \, h_{j+1}(\theta) \frac{\partial}{\partial \beta^{i-1}} g_\theta, \tag{73}$$

returns

$$\frac{\partial}{\partial \beta^{i-1}} g_\theta = -\partial_\theta \langle j_{i\theta} \rangle. \tag{74}$$

Combining all these results, we have

$$\begin{aligned}
\int_{\mathbb{R}} \mathrm{d}\eta \, \langle q_\eta \rangle \frac{\partial}{\partial \beta^{i-1}} g_{\eta+\theta} &= -\int_{\mathbb{R}} \mathrm{d}\eta \, \langle q_\eta \rangle \partial_\eta \langle j_{i,\eta+\theta} \rangle \\
&= \int_{\mathbb{R}} \mathrm{d}\eta \, \partial_\eta \langle q_\eta \rangle \langle j_{i,\eta+\theta} \rangle \\
&= -\int_{\mathbb{R}} \mathrm{d}\eta \, \frac{\partial}{\partial \beta^0} g_\eta \langle j_{i,\eta+\theta} \rangle \\
&= -\int_{\mathbb{R}} \mathrm{d}\eta \, \frac{\partial}{\partial \beta^0} g_{\eta-\theta} \langle j_{i\eta} \rangle.
\end{aligned} \tag{75}$$

Now we are ready to establish the flow equation for the free energy fluxes. Applying (69) and (75), it follows that

$$
\begin{aligned}
\frac{\partial}{\partial \beta^0} \frac{\delta g_i}{\delta \lambda_\theta} = \frac{\partial}{\partial \beta^{i-1}} \frac{\delta f}{\delta \lambda_\theta} &= \int_{\mathbb{R}} \mathrm{d}\eta \left[ \frac{\partial}{\partial \beta^{i-1}} \langle q_\eta \rangle (g_{\eta+\theta} - g_{\eta-\theta}) + \langle q_\eta \rangle \frac{\partial}{\partial \beta^{i-1}} (g_{\eta+\theta} - g_{\eta-\theta}) \right] \\
&= \int_{\mathbb{R}} \mathrm{d}\eta \left[ \frac{\partial}{\partial \beta^0} \langle j_{i,\eta} \rangle (g_{\eta+\theta} - g_{\eta-\theta}) + \langle j_{i\eta} \rangle \frac{\partial}{\partial \beta^0} (g_{\eta+\theta} - g_{\eta-\theta}) \right] \\
&= \frac{\partial}{\partial \beta^0} \int_{\mathbb{R}} \mathrm{d}\eta \langle j_{i\eta} \rangle (g_{\eta+\theta} - g_{\eta-\theta}),
\end{aligned} \tag{76}
$$

yielding

$$
\frac{\partial}{\partial \beta^0} \left( \frac{\delta g_i}{\delta \lambda_\theta} - \int_{\mathbb{R}} \mathrm{d}\eta \langle j_{i\eta} \rangle (g_{\eta+\theta} - g_{\eta-\theta}) \right) = 0. \tag{77}
$$

We expect both terms in the parenthesis to vanish in the limit of zero particles, when $\beta^0 \to -\infty$. Assuming this, we obtain the desired flow equation

$$
\frac{\delta g_i}{\delta \lambda_\theta} = \int_{\mathbb{R}} \mathrm{d}\eta \langle j_{i\eta} \rangle (g_{\eta+\theta} - g_{\eta-\theta}). \tag{78}
$$

## E   Bethe Ansatz Flow Equations

In this section we show that the flow equations obtained are the same as those obtained by varying the S-matrix in Thermodynamic Bethe Ansatz (TBA). All results for TBA in this section are found in [33].

The free energy and free energy fluxes in TBA are:

$$
f = \int d\theta \, p(\theta) \frac{\partial_\theta L(\varepsilon(\theta))}{2\pi}, \qquad g_\theta = \frac{\partial_\theta L(\varepsilon(\theta))}{2\pi}. \tag{79}
$$

Note that the free energy is obtained by integrating $g_\theta$ multiplied by $p(\theta)$. The free energy function $L(\varepsilon)$ is, for various systems:

$$
L(\varepsilon) = \begin{cases} e^{-\varepsilon} & \text{classical particles} \\ \log(1 + e^{-\varepsilon}) & \text{quantum fermions} \\ -\log(1 - e^{-\varepsilon}) & \text{quantum bosons} \end{cases} \tag{80}
$$

where the pseudo-energy solves

$$
\begin{aligned}
\varepsilon(u) &= \beta^u - \int_{\mathbb{R}} \frac{\mathrm{d}v}{2\pi} \varphi(v, u) L(v) \\
&= \beta^u - \int_{\mathbb{R}} \frac{\mathrm{d}v}{2\pi} \partial_v \phi(v, u) L(v) \\
&= \beta^u + \int_{\mathbb{R}} \frac{\mathrm{d}v}{2\pi} \phi(u, v) \partial_v L(v).
\end{aligned} \tag{81}
$$

Recall that $\beta^u$ fixes the GGE, by the weight $e^{-W}$ with $W = \beta^u Q_u$. We used $\varphi(u, v) = \mathrm{d}\phi(u, v)/\mathrm{d}u$, where the phase shift $\phi(u, v) = \phi_{uv}$ along the flow is related to the deformation parameter as $\phi_{\theta\eta} = -(\lambda_{\theta\eta} - \lambda_{\eta\theta})$. Note that the symmetry of the differential phase shift $\varphi(u, v) = \varphi(v, u)$, often seen in integrable systems, is not assumed here.

Then the derivative of $\varepsilon$ with respect to $\lambda_{\theta\eta}$ can be computed as

$$
\frac{\mathrm{d}\epsilon(u)}{\mathrm{d}\lambda_{\theta\eta}} = \int_{\mathbb{R}} \frac{\mathrm{d}v}{2\pi} \left[ -(\delta(u-\theta)\delta(v-\eta) - \delta(u-\eta)\delta(v-\theta))\partial_v L(v) + \phi(v,u)\frac{\partial}{\partial v}\frac{\partial}{\partial\lambda_{\theta\eta}}L(v) \right],
$$
$$
= \delta(u-\eta)\frac{L'(\theta)}{2\pi} - \delta(u-\theta)\frac{L'(\eta)}{2\pi} + \int_{\mathbb{R}} \frac{\mathrm{d}v}{2\pi}\varphi(v,u)n(v)\frac{\mathrm{d}\epsilon(v)}{\mathrm{d}\lambda_{\theta\eta}}, \tag{82}
$$

which implies

$$
\frac{\mathrm{d}\epsilon(u)}{\mathrm{d}\lambda_{\theta\eta}} = \frac{1}{2\pi}\left( R_{u\eta}L'(\theta) - R_{u\theta}L'(\eta) \right), \quad R_{\theta\eta} := \left( 1 - \frac{\varphi^{\mathrm{T}}}{2\pi}n \right)^{-1}_{\theta\eta}. \tag{83}
$$

We can compute the object of interest

$$
\begin{aligned}
\frac{\mathrm{d}f}{\mathrm{d}\lambda_{\theta\eta}} &= \int_{\mathbb{R}} \frac{\mathrm{d}u}{2\pi}p'(u)n(u)\frac{\mathrm{d}\epsilon(u)}{\mathrm{d}\lambda_{\theta\eta}} \\
&= \int_{\mathbb{R}} \frac{\mathrm{d}u}{2\pi}p'(u)n(u)\frac{R_{u\eta}L'(\theta) - R_{u\theta}L'(\eta)}{2\pi} \\
&= \rho(\eta)L'(\theta) - \rho(\theta)L'(\eta) \\
&= \rho(\theta)g_\eta - \rho(\eta)g_\theta,
\end{aligned} \tag{84}
$$

where $\rho(\theta) = n(\theta)\int_{\mathbb{R}}\mathrm{d}u\, R_{\theta u}p'(u)/(2\pi) = \langle q_u \rangle$ is the density of particles at rapidity $u$. Thus the TBA free energy density indeed satisfies the flow equation (18).

We next look at that for the free energy fluxes $g_i$, where almost all the above steps carry over, and we end up with

$$
\frac{\mathrm{d}g_i}{\mathrm{d}\lambda_{\theta\eta}} = \langle j_{i\theta} \rangle g_\eta - \langle j_{i\eta} \rangle g_\theta, \tag{85}
$$

which again shows that the TBA free energy fluxes satisfy the flow equation (19).

Therefore, the TBA form of the free energy density and free energy fluxes reproduces satisfy the flow equations, and hence the TBA gives the unique solution to the flow equations. This shows the TBA by direct computations without invoking integrability.

In boost symmetric cases, i.e. $\varphi(u,v) = \varphi(u-v)$, one carries out the same computation and correctly gets

$$
\begin{aligned}
\frac{\delta f}{\delta\lambda_\theta} &= \int_{\mathbb{R}} \mathrm{d}\eta\, \rho(\eta)(g_{\eta+\theta} - g_{\eta-\theta}), \\
\frac{\delta g_i}{\delta\lambda_\theta} &= \int_{\mathbb{R}} \mathrm{d}\eta\, \langle j_{i\eta} \rangle(g_{\eta+\theta} - g_{\eta-\theta}).
\end{aligned} \tag{86}
$$

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
