# Peer review of "The Space of Integrable Systems from Generalised $T\bar{T}$-Deformations"

_SciPost Physics, doi:SciPost Phys. 13, 072 (2022)_

## Round 2 · Referee Report · Anonymous (Referee 1) · 2021-8-3

Report

The work of the authors meets the SciPost publication criteria. I do recommend publication on SciPost, following some minor amendments detailed in the Requested Changes section.

Requested changes

1- In the second paragraph of Section 1, the authors cite some of the existing literature on TTbar deformations. The cite papers [8-14] following the phrase "The principles underlying TTbar-deformations have been extended to non-relativistic systems, where they are best expressed as charge-current deformations including higher-spin charges". Most of the cited works, however, deal with relativistic systems and all but two with the simple (non-generalized) TTbar deformation. Also, they cite the work [15], dealing with the standard TTbar and which appeared at the same time as [8], sparking the flurry of work on the subject, later, in a phrase referring to higher-spin TTbar models. I suggest that the authors restructure this paragraph carefully, citing the existing literature with more attention. A work dealing with non-relativistic TTbar deformations they forgot to mention is, for example, "Conserved currents and TTbar_s​ irrelevant deformations of 2D integrable field theories" by Conti, Negro and Tateo, JHEP 11 (2019) 120; e-Print: 1904.09141.

2 - In the second paragraph of Section 2, the authors display the formula $\delta H \in T\Sigma^{\textrm{Int}}$, without mentioning what $T\Sigma^{\textrm{Int}}$ is. While this might be known to readers familiar with the work [8], I think it would be worth spending a few words on its meaning.

3 - Right after, in the phrase "Because the deformations form a Hamiltonian flow [...]", they probably meant "Because $\textbf{these}$ deformations form a Hamiltonian flow [...]".

4 - In the first paragraph on page 4, 5th line, they use the acronym SM, probably meaning Standard Model. I think it is always best to avoid acronyms that have not been previously explained, as widespread as their use can be.

5 - On page 5, 2nd line after eq (5), they refer to eq (42). This is probably due to a duplicate label in the TeX file since equations (5) and (42) are identical but in different places in the work. It is better to refer to (5) here.

6 - In the last paragraph of Section 6 the authors say "The generalisation of the deformation to this situation is presented in the appendix." It would be good to specify which of the appendices they refer to.

7 - In equations (34) to (36) the authors use the function $\chi$ having an inequality as argument. They should explain what this function is.

8 - In equation (76) the authors introduce a function $\rho$. They should explain what this function is.

---

## Round 3 · Referee Report · Anonymous · 2021-11-17

Report

This work extends the notion of generalised $T\bar{T}$ deformations, including the complete set of extensive charges. They show that the deformation leads to a general deformation of the S-matrix. The authors derive flow equations to the free energy and its fluxes. Moreover, they show that the substitution of the deformed S-matrix in the TBA equation leads to the same results.

The article meets the publication criteria of SciPost Physics, and I do recommend the publication on SciPost Physics after some clarification (see Requested changes).

Requested changes

1- In the first paragraph of Section 5 the authors write the following confusing sentence: "Here we show that the generalised $T\bar{T}$-deformation provides a novel derivation of TBA", but in the Conclusion they write "We showed [...] that the thermodynamics of the deformed theories coincides with that obtained by TBA". The former sentence should be rephrased since the derivation in Appendix F starts with stating the deformed TBA equations and results in the flow equation. I also suppose that in the last paragraph of Section 5 the authors intended to refer to Appendix F instead of C.

---

## Round 3 · Referee Report · Anonymous · 2021-11-22

Strengths

- proposes an interesting generalisation of TTbar deformations
- shows the implications on the infinite-volume S-matrix
- derives flow equations for the deformed theory

Weaknesses

- imprecise on the properties of these generalised transformations
- does not compare with previously determined flow equations
- no examples or physical discussion provided

Report

Dear Editor,
this article proposes a generalisation of the current-current deformations discussed by Smirnov and Zamolodchikov (that already generalise the celebrated TTbar deformation). The authors derive the effects of these deformations on the S-matrix of the theory (in infinite volume) and write down flow equations for the charges.

The topic is interesting and some of the authors' result seem correct. However, this work needs revision in several points, which I discuss below. It is my recommendation that the paper should not be accepted for publication until such a major revision has been made.

The referee

Requested changes

1. In the introduction, the authors state that "A physical insight into TTbar was gained [...]" by relating them to changes of particle width. To put it mildly, this is a very partial statement. Physical insights in TTbar include their description as quasi-local deformations, their relation to two-dimensional gravity to string theories on the worldsheet and in target space, and to holography. The authors should mention all that.
2. In section two the authors talk about a "more judiciously chosen" set of charges. However, it becomes clear later that these charges do not necessarily satisfy physical unitarity and crossing (not to mention real / Hermitian analyticity). The authors should make it clear "what are this charges good for", see also my points 3. and 6. below.
3. Related to point 2., it would be good if the authors discussed in some detail the deformation of one simple theory (such a Sinh-Gordon), for some example of deformations that they propose that were not previously in the literature. In particular, the authors should present and discuss the finite-volume spectrum for such deformations (for instance , the kappa, eta and lambda deformations that they introduce), also as a way to put their newly-developed formalism to the test.
4. The names eta and lambda deformations, and to a lesser extent kappa, are commonly used in the literature of integrable deformations of sigma models (they are types of quantum deformations). The authors should probably pick new names.
5. In section 4 and 5 the authors discuss the flow equations and thermodynamic Bethe ansatz for their deformations. Throughout the discussion it is unclear to me whether the theory is in finite volume or at finite temperature. If I recall correctly, this made quite a difference in the authors ref. [16]. The authors should clarify this, and explain in detail whether or not their results match the one of [16] in the case where they are both applicable. This is far from immediately clear.
6. Related to point 2., I find the discussion of section 6 imprecise. The requirement of crossing symmetry and of physical unitarity (for real momenta) seem sufficient to rule out these newly-constructed deformations. These requirement are considerably weaker than imposing that the S-matrix is analytic in the whole physical strip. More generally, for two dimensional integrable QFTs there is a well define list of properties that may be demanded of the S-matrix, related to a well-defined list of physical principles: Poincare' invariance, locality, causality, unitarity, parity, time-reversal, particle-to-antiparticle symmetry, existence of bound-states. The authors should clarify which of these properties are broken by their new deformations with respect to the "usual TTbar" ones, and if possible provide example of known theories of such a type.

  • validity: good
  • significance: good
  • originality: ok
  • clarity: ok
  • formatting: excellent
  • grammar: excellent

Author:  Takato Yoshimura  on 2022-07-11  [id 2653]

(in reply to Report 2 on 2021-11-22)

  1. Thank you for the comment. Obviously, the physical interpretation in terms of particle widths is the one that is most relevant for the present paper, which is why it was mentioned. But indeed, those important aspects of $T\bar{T}$-deformations should have been mentioned. We have covered these points in the introduction of the revised version.

  2. Our motivation to introduce and use the quasi-local charges $Q_\theta$ for deforming theories stems from thermodynamics. In the context of thermalisation of isolated quantum many-body systems, it has become clear that in order to accurately describe the states to which systems relax, one has to include not only usual local conserved charges but also quasi-local ones. This is crucial both in quantum quenches, and in the hydrodynamics of integrable systems. Including quasi-local charges is in general essential in order to obtain a complete set of extensive charges. In integrable systems, the complete set of extensive charges has a basis labeled by the quasi-momentum $\theta$. Therefore it is natural to consider a deformation by these charges, which is what we do in this article.

    The referee said ``these charges do not necessarily satisfy physical unitarity and crossing (not to mention real / Hermitian analyticity)." Formally, there is no notion of crossing or real / Hermitian analyticity for conserved charges. The charges we introduce are Hermitian and extensive, and, as we infer from the physics of relaxation in integrable systems, these are all the properties that are required of conserved charges. We have added a comment on page 5 pointing out this argument, and we have clarified that in integrable systems, the choice of the charges associated to the quasi-particles of integrability are a particular case of this general argument.

    It is true, however, that the inclusion of quasi-local charges makes the analytic structure of the $S$-matrices much more intricate. Further understanding of these aspects is certainly desirable, but we believe that it is beyond the scope of the present manuscript; it is sufficient, at the level of generality that we adopt, to discuss the general picture, as we do in Section 6.

  3. Related to the point 2, we fully agree that working out a particular example would clarify some of the aspects of the generalised $T\bar{T}$-deformation proposed in this article, and in particular how it differs from the conventional ones. We however think that it would require substantial additional works and this should be better left for future studies.

    In particular, for the sinh-Gordon model as the referee suggests, the exact quasi-local charges have not been written in terms of fundamental fields; hence it is not possible, at the current stage of understanding, to write explicitly the deformation. For the result we establish, such an explicit writing is not necessary, as the result follows from general principles and the existence of the charges $Q_\theta$. However, for a more explicit example, this would be required.

    Models where explicit calculations may be done would be the XXZ model for instance, where quasi-local charges have been constructed. However, as mentioned, this would require much more work, and would be beyond the scope of this paper, where we are interested in universal features. We note in particular that the deformation is not written explicitly, as it is defined as a flow; this is sufficient for the general results we establish, but for a particular example, one would need to solve the flow, which is nontrivial.

    Further, analysing explicitly the deformed theory, even once the Hamiltonian is written explicitly, is likely to require going beyond standard methods of integrability, because of the quasi-local nature of the deformation.

    But in this paper, we concentrate on the general, universal features, not on model-specific features, which we believe are better left for future works.

    We have added comments in this respect in the introduction and conclusion. We have also added comments in the main text giving, when it is useful, the example of the sinh-Gordon model: page 4 top about the spins of local charges, and page 6, top, about how local charges deform the dispersion relation only in a restricted way.

    Concerning finite-volume spectra: it is not the goal of this paper to analyse these spectra, especially for the simpler kappa and eta deformations which have been analysed in previous literature. The point of this paper is to show that a general scattering matrix can be obtained by lambda deformation, and that the universal formalism of statistical mechanics with arbitrary set of conserved charges lead to the flow equations, and to the TBA equations in integrable models. We use finite-volume regularisation only in order to carefully study the lambda-deformations.

  4. As we do not discuss integrable deformations of any particular models, such as sigma models, and as deformations mentioned by the referee are, in our understanding, known in a somewhat restricted community (and our deformations are explicitly defined), we think there is little room for confusion; in particular experts in sigma models will easily realise that these are different deformations. So we decided not to change the names of the deformations.

  5. In this manuscript the system is in a GGE parameterised by $\beta^\theta$, which is conjugate to the quasi-local charge $Q_\theta$ in the infinite volume. Therefore, the system is in infinite volume (we have made this more precise already at the start of Section 2), and the set of states considered are not only thermal states, but the full generality of generalised Gibbs ensembles (we have clarified this at the start of Section 4). Of course, by standard arguments, systems on infinite volumes in GGE are related to the ground state energy in finite volume with special, twisted boundary conditions. But the analysis of this, and the generalisation to the excited states, would require further investigations, and is not required for our results.

    A comparison with the ref. [16] is certainly meaningful. The situation the authors deal with in there correspond to the following choice of the deformation parameter, which is nothing else than the CDD factor: $\lambda_\theta=\sum_{j: \mathrm{odd}}\alpha_je^{-j\theta}$. The GGE is parameterised as $\beta^\theta=\sum_j\beta^je^{j\theta}$. The flow equation eq (17) in their paper, which is for the ground state in the finite volume and hence can be thought of as equivalent to our flow equations, then reads, in terms of our convention, \begin{equation} \frac{\delta g_j}{\delta\alpha_n}=g_{-n}\frac{\delta g_j}{\delta\beta^n}-g_n\frac{\delta g_j}{\delta\beta^{-n}}. \end{equation} This can be recovered from eq. (18) and (19), which are the flow equations for the system with boost symmetry, in our paper. We explained how one can show that in the revised version, see Sect 4.4. 6. No, the requirement of unitarity and crossing symmetry does not rule out our deformed theories.

    First, unitarity is preserved; this was mentioned already, but we have emphasised it more, especially in section 6. Indeed, it can be immediately seen that the phase shift $\phi(\theta,\alpha)$ induced by the deformation make by definition the S-matrix unitary because $\phi(\theta,\alpha)=\lambda_{\theta\alpha}-\lambda_{\alpha\theta}=-\phi(\alpha,\theta)$.

    Second, let us remark that crossing symmetry does not have to be satisfied by the S-matrix when the deformed theory is not a relativistic QFT. As we are in general {\em not} restricting ourselves to relativistic QFT (the undeformed theory is not assumed to be relativistic, or even to be a QFT), then this does not rule out the deformed theory. Thus our deformed theories are not ruled out by this basic requirement.

    Nevertheless, it is indeed interesting to consider the case where the undeformed and deformed models are relativistic QFT. This is discussed in Section 6.

    Considering the properties mentioned by the referee: Poincar\'e invariance is obviously preserved by the deformation if $\lambda_{\theta\alpha} = \lambda_{\theta-\alpha}$. Strict locality can be broken, but this is well known already for the usual \ttbar-deformations and related to the potential lack of UV completeness (for instance, it is known that certain $T\bar{T}$-deformations give positive lengths to particles). The existence of bound states as related to the analytic properties of the S-matrix is discussed at length in section 6. New particles in the asymptotic spectrum may appear under the deformation, but the deformed S-matrix only represents particle species in direct correspondence with those of the undeformed theory. Other asymptotic particle species (such as bound state), if any, have matrix elements that can be calculated from the poles of the S-matrix by the standard techniques of QFT. Both the S-matrix, and the TBA, are well-defined quantities on the subset of particle species corresponding to the original particles. As far as we understand, causality is related to analyticity of the S-matrix, but we prefer avoiding any (rather tricky) discussion of causality and keeping with the simpler discussion of analyticity as related to the particle spectrum. Finally, with this partial knowledge of the S-matrix, it is not possible to fully address CPT and crossing symmetry. Indeed, CPT and crossing symmetry require the knowledge of charge-conjugated particles, which we may not know because, as mentioned, of potentially new particles in the asymptotic spectrum.

---

## Round 3 · Author Response

We thank the reviewer for his/her positive assessment. We have implemented several changes in response to the referee's each comment.

---

## Round 3 · List of Changes

1. We rearranged the way we cite these references so that it's more accurate. We also cited the paper mentioned by the referee as well as another article on non-relativistic $T\Bar{T}$-deformations.
2. We added the definition of $T\Sigma^\mathrm{Int}$.
3. We corrected the typo.
4. We meant ``supplement material" by SM. We rephrased it as the appendix.
5. The reference to eq (5) is corrected.
6. We added the reference to a particular section in the appendices.
7. The definition of the indicator function $\chi$ is added.
8. The definition of $\rho(\theta)$ is added.

Resubmission 2105.03326v4 on 30 December 2021

---

## Round 4 · Author Response

We are grateful for the reviewer's constructive suggestion. It was indeed a little unclear what we meant by "derivation of TBA". The point here is that because we already argued in the previous sections that a generalised TTbar-deformed theory is an integrable system whose phase shift is $\lambda_{\theta\phi}$, the solution of the flow equations should naturally describe the thermodynamics of that integrable system. Now, what is shown in Appendix F is that the free energy as well as the free energy fluxes of an integrable system with the phase shift $\lambda_{\theta\phi}$, obtained from TBA, indeed satisfy the flow equations with an appropriate free theory intial condition ( $\lambda_{\theta\phi}=0$). Since these equations form a closed first-order system, the smooth solution must be unique. Hence this serves as a novel confirmation/derivation of TBA.

---

## Round 4 · List of Changes

1. We added the following sentence in Sect. 5: "To be more precise, we shall show that by solving the flowequations of a generalised $T\bar{T}$-deformed theory parameterised by $\lambda_{\theta\phi}$, which was shown to be equivalent to the integrable model whose phase shift is $\lambda_{\theta\phi}$ in Sect. 3, we obtain the free energy and the free energy fluxes that coincide with those of the corresponding integrable.

  2. We replaced Appendic C with F in the last paragraph of Section 5.

---

## Round 5 · Referee Report · Anonymous (Referee 3) · 2022-8-10

Report

Dear Editor,
the authors have addressed the comments raised in the previous report in a satisfactory way. I think that the manuscript, in its current form, is suitable for publication.
The referee

---

## Round 5 · Author Response

Dear Editor,

We are grateful for your considerations and thank the reviewer for his/her positive assessment.

Some important points have been raised by the referee. We answer all points below, and spell out the changes made in response to the referee's each comment.

We have also improved the introduction, in order to better emphasise the ideas and results. We have improved Section 4 by bringing in the appendix on the commutativity of deformed conserved charges, as this is an essential element of the theory which is not too technically complicated, and also by making subdivision in order to clarify the logic. We have also included there an explicit comparison with the results of ref [16]. We have improved Appendix B about the calculation of the scattering phase using classical particle systems. We have also made additional small improvements to clarify the explanations throughout the text.

We point out, in particular, that the paper not only proposes a much wider family of TTbar-deformations, with consequences on generating integrable models with arbitrary scattering, but also provides a new and universal derivation of the flow equations, not only for the free energy but also for the free energy fluxes. This is done within a formulation of statistical mechanics that accounts for an arbitrary number of extensive conservation laws, and gives rise, for instance, to a new derivation of the TBA that does not require the Bethe ansatz -- thus valid in both quantum and classical systems.

Although the paper stays mostly at the formal level, we believe the ideas introduced are particularly important, and have strong potential for further development. For instance, we believe it is important to emphasise the existence and consequences of quasi-local charges, and the strength of the associated universal formalism for statistical mechanics which accounts for these -- something which has not been appreciated until now beyond the community working on non-equilibrium many-body systems, and in particular not in the community working on TTbar-deformations. The ideas and methods we use and introduce are novel, and, we believe, very much worth disseminating.

---

## Round 5 · List of Changes

1. We included the physical interpretations of the TTbar-deformation that were not mentioned in the manuscript before.

  2. We explained why it is important and meaningful to consider the deformations by quasi-local charges from the statistical-mechanical point of view.

  3. We commented on why it is in general hard to work out a concrete example for the generalised TTbar-deformation in Introduction and Conclusion. We have also added comments in the main text giving, when it is useful, the example of the sinh-Gordon model: page 4 top about the spins of local charges, and page 6, top, about how local charges deform the dispersion relation only in a restricted way.

  4. A comparison with the ref. [16] and in particular the derivation of the flow equation obtained in there within our formalism is added.

  5. We elucidated which properties of the S-matrices that are usually expected to hold in relativistic QFTs remain true upon the deformations. We also clarified that to establish crossing symmetry for the S-matrices requires further understanding of the deformations, which for the moment is out of reach.

---

## Editorial Decision

published